# MED-REFL: ENHANCING COMPLEX REASONING VIA FINE-GRAINED SELF-CORRECTION

## ABSTRACT

Large reasoning models excel in domains like mathematics where intermediate reasoning is straightforward to verify, but struggle to self-correct in medicine fields where evaluating intermediate reasoning is cumbersome and expensive. This verification bottleneck hinders the development of reliable AI reasoners for high-stakes application. Here we propose **Med-REFL**, a novel framework that learns fine-grained reflection without human labels or model distillation. Med-REFL introduces a deterministic structural assessment of the reasoning space to automatically generate preference data for reflection. By globally evaluating all explored reasoning paths in a tree-of-thoughts, our method quantifies the value of corrective actions, enabling the automated construction of direct preference optimization pairs. This trains the model to recognize and amend its own reasoning fallacies. Extensive experiments show Med-REFL delivers robust gains across diverse models architectures and medical benchmarks, boosting a general-purpose Llama3.1-8B by +5.82% and the state-of-the-art Huatuo-o1 by +4.13% on the MedQA benchmark. Our Med-REFL-8B achieves state-of-the-art performance among 7-8B models while even competing with models twice its size. Crucially, targeted ablations prove its success generalizes to other domains such as logical reasoning and mitigates the 'fake reflection' phenomenon in LRMs. Ultimately, our framework provides a scalable solution to the verification bottleneck, paving the way for more reliable AI reasoners in high-stakes domains like medicine. Med-REFL has been made publicly available [1].

## 1 INTRODUCTION

Recent advancements in large reasoning models (LRMs) have demonstrated remarkable progress in domains like mathematics and code generation (Plaat et al., 2024). A key characteristic of these fields is that their intermediate reasoning steps adhere to clear, objective, and easily verifiable standards of correctness, which provide strong supervisory signals for effective model training (Hui et al., 2024; Li et al., 2025). However, this success has not seamlessly transferred to knowledge-intensive and high-stakes domains such as medicine (Hoyt et al., 2025; Huang et al., 2025). The core challenge lies in the unique complexity of medical reasoning: the evaluation of intermediate steps is cumbersome and expensive, fraught with profound uncertainty, and lacks cheap, reliable verification methods (Jiang et al., 2025; Wu et al., 2025; Qiu et al., 2025). This verification bottleneck is the critical reason why methods effective in mathematics falter in the medical domain. It prevents models from effectively learning and reflecting on their errors—a crucial process for improving reasoning ability (Zhang et al., 2024a; Kamoi et al., 2024; Xi et al., 2024).

Despite these challenges, previous attempts to enhance medical reasoning have made valuable progress. The first paradigm, relying on supervised fine-tuning (SFT) (Dong et al., 2024) with data distilled from costly teacher models or on outcome-based reinforcement learning (RL) (Cheng et al., 2023; Dai et al., 2024), is inherently result-oriented. These methods primarily rely on final answers as selection or reward signals, paying insufficient attention to intermediate processes (Xi et al., 2024). This makes distillation methods prone to learning flawed logic that coincidentally produces correct answers, while outcome-based RL is vulnerable to reward hacking, both failing to guarantee robust reasoning (Miao et al., 2024; Bai et al., 2022). The second paradigm, which leverages process

---

[1] Details can be found in Appendix C.

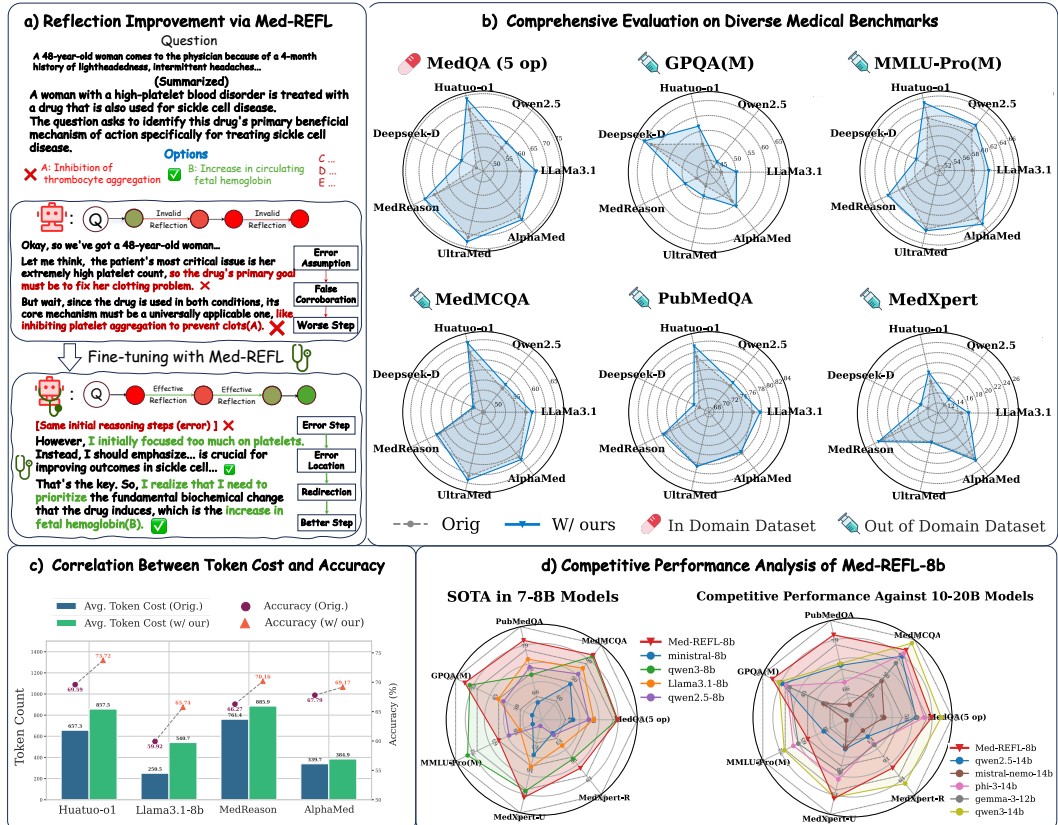

Figure 1: Qualitative and quantitative analysis of Med-REFL's effectiveness. a) An example illustrating the change in Huatuo-o1's reflection pattern. Before fine-tuning, the model follows an invalid reflection path based on an error assumption and false correlation. After fine-tuning with Med-REFL, it demonstrates effective reflection by locating the reasoning error and redirecting its path to the correct answer. b) Comprehensive performance evaluation on diverse medical benchmarks, showing consistent accuracy improvements (w/ ours) over the original models (Orig). Deepseek-D refers to Deepseek-Distill-8B. c) The relationship between average token cost per question and performance on MedQA(5 op). It suggests a link between reasoning length and correctness. d) Competitive analysis of our Med-REFL-8B model. The left chart shows it outperforms most models in the 7-8B class. The right chart demonstrates its high efficiency, achieving performance comparable to or exceeding much larger 10-20B models.

reward models (PRMs) (Li & Li, 2025), provides more granular stepwise feedback. However, this approach often necessitates training a separate, demanding reward model and creates a complex, multi-stage pipeline (Zhang et al., 2025c). More critically, this paradigm is fundamentally designed to identify and reward the most correct path, which does not effectively teach the model the essential skill of how to reflect on and recover from its own diverse errors (Madaan et al., 2023).

To overcome these limitations, we introduce **Med-REFL**, a novel framework that learns fine-grained reflection within a unified, resource-efficient system designed to specifically address the verification bottleneck in complex medical reasoning. Our approach begins with a deterministic structural assessment of the reasoning space, using a tree-of-thoughts (ToT) (Long, 2023; Yao et al., 2023) to map out diverse reasoning pathways. This structured map is the key to mitigating the 'reward hacking' problem; by aggregating outcomes over the entire tree, we derive a robust quality score, step value, for each intermediate state. To enable the model to learn self-correction, we further introduce our pivotal metric, the action value, which utilizes the tree's global information to evaluate corrective actions. This metric measures how much these actions improve the entire problem-solving trajectory, thereby guiding the model's reflective reasoning learning. Action value guides the construction of high-quality preference data for direct preference optimization (DPO) (Rafailov et al., 2023), en-

Table 1: Comparison of different LLM reasoning strategies based on their core attributes. Med-RELF addresses the limitations of previous approaches. Expert refers to closed-source teacher model or human experts.

| Method | Medical Specific | Reflect Ability | Process Supervision | Expert Free | PRM Free | RAG Free |
|--------|:---:|:---:|:---:|:---:|:---:|:---:|
| Huatuo-o1 (Chen et al., 2025) | ✓ | ✓ | ✗ | ✗ | ✓ | ✓ |
| MedReason (Wu et al., 2025) | ✓ | ✗ | ✓ | ✗ | ✓ | ✗ |
| O1-Journy (Huang et al., 2024b) | ✗ | ✓ | ✗ | ✗ | ✓ | ✓ |
| AlphaMed (Liu et al., 2025) | ✓ | ✗ | ✗ | ✓ | ✓ | ✓ |
| Med-PRM (Yun et al., 2025) | ✓ | ✗ | ✓ | ✗ | ✗ | ✗ |
| Med$^3$ (Jiang et al., 2025) | ✓ | ✗ | ✓ | ✓ | ✗ | ✓ |
| Med-RLF(ours) | ✓ | ✓ | ✓ | ✓ | ✓ | ✓ |

abling a contrast not just between good and bad states, but more importantly, between invalid and effective reflection as shown in Figure 1 a). This process directly instills the capability for genuine self-correction.

Our extensive experiments validate this approach from multiple perspectives. **First**, as detail in Table 2, Med-REFL demonstrates remarkable versatility by enhancing models across different post-training paradigms. It yields the most substantial improvements on reason-heavy models, with an average gain of +3.4% at Huatuo-o1-8B (Chen et al., 2025), pushing already state-of-the-art (SOTA) models to new performance heights. **Second**, in Figure 1 b), the framework shows robust generalization, delivering a significant +3.67% average gain on the in-distribution MedQA (Jin et al., 2021)) benchmark while maintaining strong performance across a range of challenging out-of-distribution(OoD) datasets like GPQA (+3.53%) (Rein et al., 2024). **Third**, As shown in Figure 1 d), our Med-REFL-8B model not only achieves SOTA performance in the 7-8B models but also demonstrates strong competitiveness that rivals or even surpasses models in the 10-20B class. Crucially, targeted ablations not only confirm its architectural superiority over statistical sampling approaches, but also reveal that Med-REFL instills a genuine, activatable self-correction capability that generalizes to other domains (e.g., open-ended logic problems) and provides a solution to 'fake reflection' phenomenon in LRMs.

Our main contributions are as follows:

1. We propose Med-REFL a novel framework to overcome the critical bottleneck of expensive intermediate-step verification in medical reasoning. To our knowledge, our approach is the first to successfully enhance complex reasoning through focusing on improving fine-grained reflection.

2. We introduce a deterministic, structure-based evaluation that yields an experimentally-validated learning signal, enabling the automatic generation of fine-grained reflection data and overcoming the limitations of both outcome-oriented learning and PRMs.

3. We demonstrate through extensive experiments that Med-REFL consistently improves performance across various models and benchmarks, and is capable of training new SOTA models. Via targeted ablations, we further prove that this success stems from internalized self-correction, which effectively mitigates the pervasive fake reflection in LRMs.

## 2 MED-REFL OVERVIEW

Our methodology is designed to teach the model a instilled reflection capability. The pipeline, illustrated in Figure 2, comprises four main stages: (1) generating a diverse map of reasoning trajectories ToT approach; (2) establishing a deterministic, structure-based method to quantitatively evaluate the quality of any intermediate step and any corrective action; (3) leveraging these metrics to automatically construct two types of preference pairs—a primary set for 'Reflection Learning' and a supplementary set for 'Reasoning Enhancement'; and (4) fine-tuning the model using DPO to instill a robust self-correction capability. The comparison of different reasoning enhancement strategies can be found in Table 1.

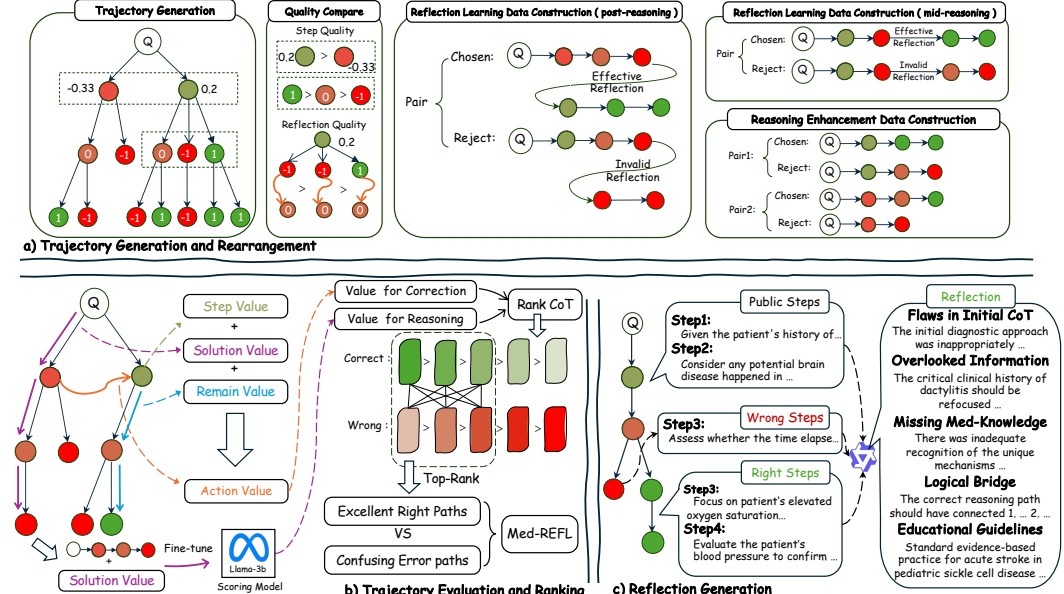

Figure 2: Overview of the Med-REFL 'reflection learning' pipeline. a) Illustrate how step value pointed and the target data pair construction in Med-REFL. b) Illustrate how action value calculated and how to rank reasoning path. c) An illustrative example showcases the fine-grained self-correction mechanism where initial reasoning flaws are identified and rectified during medical question answering.

## 2.1 TRAJECTORY GENERATION AND EVALUATION

To generate diverse reasoning trajectories, each node represents an intermediate reasoning state, and a path from root to leaf forms a complete chain of thought (CoT) trajectory. ToT (details in Appendix E.1) explore multiple reasoning directions at each step to generate diverse and high-quality trajectories.

To evaluate trajectory quality, we compare the final answer of each path to the ground truth, assigning '+1' to steps in correct paths and '-1' to steps in incorrect paths. For a step $s$, we define its quality as:

$$v_{\text{step}}(s) = \frac{N_{s,\text{correct}} - N_{s,\text{incorrect}}}{N_s}, \tag{1}$$

where $N_s$ is the number of paths in the subtree rooted at $s$, and $N_{s,\text{correct}}$ and $N_{s,\text{incorrect}}$ are the counts of correct and incorrect paths, respectively. For a trajectory $P = (s_1, \ldots, s_L)$ of length $L$, we compute its solution quality:

$$v_{\text{sol}}(P) = \frac{1}{L} \sum_{i=1}^{L} v_{\text{step}}(s_i). \tag{2}$$

Similarly, for a partial trajectory starting at step $j$, the remaining quality is:

$$v_{\text{rem}}(P') = \frac{1}{L - j + 1} \sum_{i=j}^{L} v_{\text{step}}(s_i). \tag{3}$$

To assess reflection actions, we define the action value for transitioning from node $s$ to $s'$ via action $a$:

$$v_{\text{act}}(a_{(s,s')}) = \lambda_1(v_{\text{step}}(s) - v_{\text{step}}(s')) + \lambda_2(v_{\text{sol}}(s) - v_{\text{sol}}(s')) + \lambda_3(v_{\text{rem}}(s) - v_{\text{rem}}(s')), \tag{4}$$

where $\lambda_1, \lambda_2, \lambda_3$ are tunable weights.

## 2.2 PREFERENCE PAIR CONSTRUCTION FOR REFLECTION LEARNING

To train large language model (LLM) for error correction, we construct DPO pairs that favor trajectories where reflection corrects errors over those where reflection fails. For a medical question $q$ with ground truth $y$ and its candidate CoT trajectory $P$, an error locator and a reflector are driven by pre-trained LLM $\pi$. This process uses two LLM-based tools (all prompts can be found in Appendix H):

- **Error Locator ($E_\pi$):** Identifies the first incorrect step $s_{\text{err}}$ in an erroneous CoT trajectory $P_{\text{err}}$ with it's candidate node set, $S_{\text{cand}}$, formed by the $k$ lowest $v_{\text{step}}$ score nodes:

$$s_{\text{err}} = E_\pi(q, y, S_{\text{cand}}(k), P_{\text{err}}). \tag{5}$$

- **Reflector ($C_\pi$):** Generates reflection text $R$ explaining the transition from erroneous steps $S_{\text{orig}}$ to corrected steps $S_{\text{new}}$, given common prefix steps $S_{\text{pub}}$:

$$R = C_\pi(q, S_{\text{pub}}, S_{\text{orig}}, S_{\text{new}}). \tag{6}$$

Starting from the parent node of $s_{\text{err}}$, the model explores alternative reasoning branches, classifying them as correct or incorrect based on their final answers. Next, we construct algorithms to artificially simulate two scenarios of reflection in reasoning:

- Firstly, we simulate scenarios where the model, upon encountering a challenge or **detecting an error during the reasoning process (mid-reasoning)** to correct a single erroneous step $s_{\text{err}}$: $R = C_\pi(q, S_{\text{pub}}, s_{\text{err}}, S_{\text{new}})$, with the trajectory: $P_{\text{refl}} = S_{\text{pub}} \rightarrow s_{\text{err}} \rightarrow R \rightarrow S_{\text{new}}$.

- In the second, we simulate the **reflection occurs post-reasoning**, regarding as a review after the entire erroneous trajectory from $s_{\text{err}}$ to the trajectory's end ($S_{\text{orig}}$): $R' = C_\pi(q, S_{\text{pub}}, S_{\text{orig}}, S_{\text{new}})$, with the trajectory: $P'_{\text{refl}} = S_{\text{pub}} \rightarrow S_{\text{orig}} \rightarrow R' \rightarrow S_{\text{new}}$.

- Finally, trajectories are ranked by $v_{\text{act}}$ (Equation 4), and we select a top-ranked effective reflection trajectory ($P_{\text{chosen}}$) and a top-ranked invalid reflection trajectory (reflect but lead to another wrong step) ($P_{\text{rejected}}$) as seed trajectories to construct the first-person reasoning trace $r$:

$$\begin{cases} r_{chosen} = M_\pi(P_{chosen}), \\ r_{rejected} = M_\pi(P_{rejected}), \end{cases} \tag{7}$$

where $M_\pi$ is a thinker to transform $P_{(ref)}$ or $P'_{\text{refl}}$ into a verbalized reasoning trace $r$ by reformulating the explicit steps in P into natural language thinking.

So the high-quality reasoning trace $r_{chosen}$ and error-prone reasoning $r_{rejected}$ could be generated by $M_\pi$ from $P_{\text{chosen}}$ and $P_{\text{rejected}}$ to form the DPO pair ($r_{\text{chosen}}, r_{\text{rejected}}$).

## 2.3 PREFERENCE PAIR CONSTRUCTION FOR REASONING ENHANCEMENT

To maintain broad reasoning proficiency, we supplement the reflection data with general-purpose reasoning pairs by comparing high-quality correct CoT trajectories with plausible but incorrect ones. In order to better distinguish the quality of trajectories, we train a scoring model $\pi'$ (Llama3.2-3B) with a SFT dataset $D_{\text{SFT}} = (q, P, v_{\text{sol}(P)})$. We use $\pi'$ to score the generated data to obtain a higher-quality signal $V_{\text{CoT}}(P) = \pi'(P)$. Then, trajectories are ranked by $V_{\text{CoT}}$, and similar to Equation 7, we select a high-quality correct trajectory ($P'_{\text{chosen}}$) and a plausible incorrect trajectory ($P'_{\text{rejected}}$) to generate the reasoning trace $r'$ using $M_\pi$ resulting in the DPO pair ($r'_{\text{chosen}}, r'_{\text{rejected}}$).

## 2.4 MODEL FINE-TUNING WITH DPO

After the previous three-stage process, we obtain the 33k DPO data: $D^+_{\text{DPO}} = (\{r_{\text{chosen}}, r_{\text{reject}}\}, \{r'_{\text{chosen}}, r'_{\text{reject}}\})$. Then we fine-tune LLMs with $D^+_{\text{DPO}}$ to prioritize effective reasoning and reflection patterns. This process optimizes the model to favor effective reflection instead of invalid reflection, enabling robust error correction in medical question answering.

Table 2: A comprehensive evaluation of models with different post-training methods. The table shows original accuracy (%, orig.) and the performance gain from Med-REFL (w/ ours) across various in-distribution and OoD medical benchmarks.

| Post-Training Method | | Mix | | SFT | | | | | | GRPO | | |
|---|---|---|---|---|---|---|---|---|---|---|---|---|
| Model Type | | Instruction-Tuned | | Reason-Heavy | | Knowledge-Heavy | | | | Pure-RL | | |
| Domain | Dataset | LLaMa3.1-8B | Qwen2.5-7B | Huatuo-o1-8B | Deepseek-Distill-8B | MedReason-8B | UltraMedical3.1-8B | AlphaMed-8B | Avg. |
| | | orig. w/ ours | orig. w/ ours | orig. w/ ours | orig. w/ ours | orig. w/ ours | orig. w/ ours | orig. w/ ours | |
| **In-Distribution** | MedQA (5 op) | 59.92 65.74 | 57.11 59.70 | 69.59 **73.72** | 48.85 55.00 | 66.27 70.16 | 71.34 73.08 | 67.79 69.17 | |
| | Gain | +5.82 | +2.59 | +4.13 | **+6.15** | +3.89 | +1.74 | +1.38 | **+3.67** |
| **OoD** | MedMCQA | 57.61 59.11 | 54.52 55.79 | 62.13 **64.66** | 49.51 49.83 | 58.98 59.78 | 63.30 64.31 | 61.22 62.03 | |
| | Gain | +1.50 | +1.27 | **+2.53** | +0.32 | +0.80 | +1.01 | +0.81 | +1.18 |
| | GPQA(M) | 45.16 50.22 | 43.34 45.36 | 50.67 56.80 | 62.43 **65.04** | 45.64 49.84 | 45.76 49.20 | 53.40 55.67 | |
| | Gain | +5.06 | +2.02 | **+6.13** | +2.61 | +4.20 | +3.44 | +2.27 | +3.53 |
| | MedXpert-R | 14.14 15.52 | 12.45 13.66 | 16.85 17.78 | 11.98 13.38 | 22.31 22.34 | 15.87 16.03 | 21.91 **22.43** | |
| | Gain | +1.38 | +1.21 | +0.93 | **+1.40** | +0.03 | +0.16 | +0.52 | +0.80 |
| | MedXpert-U | 16.11 17.92 | 10.79 12.84 | 15.31 20.02 | 13.07 14.92 | 22.55 25.36 | 15.68 15.85 | 20.24 **20.78** | |
| | Gain | +1.81 | +2.05 | **+4.71** | +1.85 | +2.81 | +0.17 | +0.54 | +1.99 |
| | MMLU-Pro(M) | 57.56 60.89 | 61.36 62.66 | 61.87 **64.97** | 53.83 56.23 | 59.14 62.51 | 63.06 63.43 | 63.29 64.81 | |
| | Gain | +3.33 | +1.30 | +3.10 | +2.40 | **+3.37** | +0.37 | +1.52 | +2.20 |
| | PubMedQA | 76.26 77.73 | 74.12 74.97 | 79.02 **81.32** | 69.30 70.60 | 77.17 77.69 | 78.60 78.74 | 78.03 77.70 | |
| | Gain | +1.47 | +0.85 | **+2.30** | +1.30 | +0.52 | +0.14 | -0.33 | +0.89 |
| **Average** | Accuracy | 46.68 49.59 | 44.81 46.42 | 50.78 **54.18** | 44.14 46.43 | 50.29 52.52 | 50.52 51.52 | 51.98 52.94 | |
| | Gain | +2.91 | +1.61 | **+3.40** | +2.29 | +2.23 | +1.00 | +0.96 | +2.06 |

# 3 MAIN EXPERIMENTS

## 3.1 EXPERIMENTAL SETUP

**Benchmark.** Our primary dataset for developing and evaluating Med-REFL is the MedQA-USMLE (5 options) (Jin et al., 2021) question-answering task. To further assess the generalization ability of models fine-tuned with Med-REFL, we conducted evaluations on several additional diverse medical benchmarks, including MedMCQA (validation set) (Pal et al., 2022), PubMedQA (test set) (Jin et al., 2019), GPQA medical subset (GPQA-M) and MMLU-Pro Medical subset (health and biology tracks, donate as 'MMLU-Pro(M)') (Wang et al., 2024). We also conducted tests on the expert-level benchmark MedXpertQA (Zuo et al., 2025), divided into two parts: Understanding (MedXpert-U) and Reasoning (MedXpert-R) , which are used to separately evaluate the improvement of our method on different types of tasks.

**Models, Evaluation and Training.** We evaluated Med-REFL's ability to enhance reasoning capabilities across a diverse suite of six 7B-8B parameter baseline models. To better understand its impact, we grouped these models into three categories based on their primary architectural strengths and training focus. First, the **Instruction-Tuned** models (Llama3.1-8B (Grattafiori et al., 2024) and Qwen2.5-7B (Qwen et al., 2025)) are selected to assess our framework's effectiveness at instilling specialized skills from a generalist base. Second, the **Reason-Heavy** models (Huatuo-o1-8B and DeepSeek-Distill-8B (DeepSeek-AI et al., 2025)), are specifically optimized for complex logical deduction, allowing us to test Med-REFL's capacity to refine already strong reasoning faculties. Third, the **Knowledge-Heavy** models (MedReason-8B (Wu et al., 2025) and UltraMedical3.1-8B (Zhang et al., 2024b)) are extensively fine-tuned on vast medical corpora, which are to evaluate the framework's impact on models rich in domain-specific knowledge. In addition to these SFT-trained models, the model AlphaMed (Liu et al., 2025), trained entirely with GRPO (Shao et al., 2024), was also tested. To ensure stable and reliable results, all reported scores are the average of three independent runs. More evaluation and training details could be found in Appendix E.3.

**Training Dataset Construction.** Our preference dataset was constructed from the MedQA-USMLE training set via a two-stage automated pipeline. A detailed description of the entire data

construction process, including the creation of different preference pair types, is available in Appendix E.2.

## 3.2 MAIN RESULTS AND ANALYSIS

The comprehensive results presented in Table 2 validate the potent efficacy of our Med-REFL framework. Our methodology delivers consistent and significant performance improvements across both the in-distribution MedQA benchmark and a diverse suite of OoD challenges. Notably, Med-REFL substantially elevates the performance of already SOTA specialized models (Huatuo-o1-8B, Avg. +3.40%), demonstrating its value not just as a training method, but as a powerful enhancement framework for even the strongest existing models.

**Analysis across Model Architectures.** A deeper analysis reveals a compelling pattern in how Med-REFL interacts with different model types. While it successfully instills complex reasoning skills in general-purpose instruction-tuned models and helps knowledge-heavy models better apply their vast knowledge, the most pronounced average gains are observed in the **Reason-Heavy** category like Huatuo-o1. By observing the different reasoning pattern in model's output, we hypothesize this is because these models, through their original training, have already developed a rudimentary framework for reflection. Med-REFL, in this context, acts as a powerful catalyst; it **hones and refines this pre-existing, latent capability**, leading to outsized performance improvements. Besides, it also proves its compatibility with pure reinforcement learning by consistently boosting AlphaMed's performance (e.g., +2.27% on GPQA-Med and +6.15% on MedQA.

**Performance across Benchmark Characteristics.** The framework's impact is particularly significant on datasets that demand deep, multi-step deductive reasoning, such as GPQA (Med+) and MMLU-Pro (Med+), which saw average gains of +3.53% and +2.20% respectively across all models. This demonstrates that Med-REFL directly enhances the core logical faculties of the models. Furthermore, it showcases consistent gains on the MedXpert benchmark on the most challenging dataset, MedXpertQA and the improvement for understanding questions is significantly greater than for reasoning questions. In contrast, while still positive, the gains on datasets that are more retrieval-oriented, like PubMedQA, are more modest. This is expected, as such tasks rely less on the iterative, complex self-correction process that our framework is designed to teach, further validating that Med-REFL's primary contribution is to the quality of the reasoning process itself.

**Analysis of Reasoning Depth and Competitive Performance.** We further analyzed the impact of Med-REFL on the nature of the reasoning process itself. As illustrated in Figure 1 c), it reveals a strong positive correlation between the length of a model's reasoning trace and its final accuracy on MedQA. The application of Med-REFL consistently led to longer, more detailed reasoning, which is structurally linked to the performance gains. Furthermore, Figure 1 d) highlights the exceptional standing of our Med-REFL-8B model (Huatuo-o1 trained with Med-REFL). This proves that our framework provides a resource-efficient pathway to achieving top-tier medical reasoning capabilities without relying on massive model scale. Detailed results are available in Appendix E.4.

## 4 ABLATION STUDIES AND FURTHER ANALYSIS

### 4.1 VERIFYING THE INTERNALIZATION OF REFLECTION CAPABILITY

A notable pattern emerged from our main results: Med-REFL's performance gains were most pronounced on reason-heavy models. We hypothesized that this is because such models have already developed a rudimentary 'reflection' paradigm in their post-training. Med-REFL may not just instill this capability from scratch, but act as a powerful enhancer for models that already know how to reflect, even if imperfectly. This led us to a critical hypothesis, **if the benefit of our reflection-specific training**

Table 3: Impact of different prompting strategies on Llama3.1-8B's reflection capability. (w/o RC means only us 'Reason Enhancement' data

| Model Configuration (Llama3.1-8B) | Think w/o reflect | Reflect after think |
|---|---|---|
| Base Model | 59.92 | 61.45(+1.53) |
| +Med-REFL (w/o RC) | 65.17 | 65.87(+0.70) |
| +Med-REFL (Full) | 65.74 | **68.15**(+2.41) |

**could be unlocked by trained reasoning pattern, then they should also be elicitable through prompting**. To test this hypothesis, we designed a controlled experiment on MedQA to evaluate our fine-tuned Llama3.1-8B model under two distinct prompting conditions: a standard prompt instructing it to 'think step-by-step', and a second prompt that explicitly added a 'critical reflection' stage after the initial thinking. The results, detailed in Table 3, are revealing. Under standard prompting, the full Med-REFL model already shows a clear advantage. However, when reflection is explicitly activated with the "Reflect after think" prompt, the performance of the model trained on our full dataset surges by a significant +2.41%. In contrast, the base model and the model trained without our reflection-correction (RC) data saw much smaller gains from the same prompt. This outcome provides powerful evidence for our hypothesis. It confirms that Med-REFL does not merely teach models to mimic a reflection-like writing style; it instills a genuine, internalized cognitive capability for self-correction. This capability can remain latent under standard conditions but can be explicitly activated when required, demonstrating a deeper and more robust form of learning. Another experiment in Appendix F.7 also provides evidence for this.

## 4.2 MECHANISTIC ANALYSIS: OVERCOMING FAKE REFLECTION

A critical limitation in current reasoning models is 'fake reflection' or confirmatory bias, where models predominantly reaffirm initial errors rather than genuinely correcting them(Kang et al., 2025).To verify whether Med-REFL functionally breaks this confirmation loop or merely induces stochastic instability, we utilized the trajectories from Table 3 employed GPT-4o-as-judge to evaluate answer

Table 4: Quantitative analysis of the intermediate reflection process (reflection effectiveness) on Llama3.1-8B.

| Metric | Base Model | +Med-REFL |
|---|---|---|
| Reflection Divergence Rate $(A_{think} \neq A_{reflect})$ | 1.38% | **2.23%** |
| Correction Success Rate $(A_{reflect} = Correct \mid Changed)$ | 12.76% | **19.83%** |

correctness between the initial 'Thinking' ($A_{think}$) and the 'Critical Reflection' ($A_{reflect}$) phase. We defined two key dynamic metrics to open the 'black box' of the correction: **reflection divergence rate**, measuring how often the model alters its decision after reflecting, and **correction success rate**, measuring the conditional probability that an answer change successfully rectifies an error. As illustrated in Table 4, Med-REFL increases the divergence rate from 1.38% to 2.23%, indicating that our training effectively overcomes inherent reasoning inertia to encourage active self-scrutiny. Crucially, this increased divergence is not mere noise; the **correction success rate** nearly doubles from 12.76% to 19.83%. This result demonstrates that the internalized self-reflection skills instilled by Med-REFL effectively mitigating the limitation of the LRM's 'fake reflection'.

## 4.3 GENERALIZATION ANALYSIS: ADAPTING TO NON-MEDICAL OPEN-ENDED PROBLEMS

To investigate the generalizability of our framework beyond the medical domain and constrained multiple-choice formats, we extended our evaluation to the Knights and Knaves (K&K) benchmark (Xie et al.), a challenging open-ended logic puzzle task where reasoning difficulty scales exponentially with the number of participants ($n$). We constructed a special-

Table 5: Performance on the open-ended logic puzzle benchmark (K&K). +*Puzzle-REFL* uses logic data.

| Setting | Llama3.1-8b | +Med-REFL | +Puzzle-REFL |
|---|---|---|---|
| $n = 2$ | 38.77 | 40.82 | **45.73** |
| $n = 3$ | 22.88 | 29.23 | **36.24** |
| $n = 4$ | 13.40 | 13.96 | **21.85** |
| **Avg** | 25.02 | 28.00 | **34.61** |

ized 'Puzzle-REFL' dataset by seamlessly transferring our ToT framework: using only 100 simple seed questions ($n = 2$) and adapting the prompts from medical experts to logic solvers, we efficiently generated 5,654 reflection-oriented DPO pairs. The results, presented in Table 5, are compelling. Remarkably, the model trained solely on medical data (+Med-REFL) achieved a clear performance gain over the base Llama3.1-8B (28.00% vs. 25.02%), suggesting that the self-correction capability instilled by our framework is a transferable cognitive skill rather than mere domain knowledge memorization. Furthermore, the model fine-tuned on the generated Puzzle-REFL data demonstrated superior performance with an average accuracy of 34.61%, achieving substantial improvements even on the more complex, unseen $n = 4$ puzzles (+8.45%). This empirically confirms that Med-REFL is a domain-agnostic data engineering framework capable of unlocking complex, open-ended reasoning capabilities with minimal initial resources.

## 4.4 COMPUTATIONAL COST AND EFFICIENCY ANALYSIS

Beyond performance gains, economic efficiency is also critical for medical LRM. Conventional approaches often incur prohibitive costs: training effective PRMs typically necessitates training separate, parameter-heavy reward models consuming hundreds of GPU hours, while distillation from expert model like GPT-5 involves massive API consumption. In stark contrast, Med-REFL offers a transparent

Table 6: Cost comparison between different reasoning enhancement paradigms. Estimated costs are based on standard A100 cloud rental rates ($\frac{\$2}{\text{GPU} \times \text{hour}}$) or public API pricing.

| Method | Resource Dependency | Est. Time / Cost |
|---|---|---|
| Distillation (GPT-5) | Paid API | >$2,500 (Token cost) |
| Training PRM | Reward Model | >100 GPU Hours |
| **Med-REFL (Data Gen)** | **Self-Generated** | **≈83 GPU Hours (≈$166)** |
| **Med-REFL (Training)** | **Standard DPO** | **≈12 GPU Hours** |

and highly cost-effective solution, as detailed in Table 6. our entire data generation pipeline comprises ToT rollout (≈70h), error location (≈3h), and reflection generation (≈10h) on a single A100, which only costs roughly $166 based on standard cloud rates.

## 4.5 COMPARISON WITH ALTERNATIVE STRATEGIES.

A central claim of our work is that Med-REFL's deterministic, structure-based evaluation provides a higher-quality learning signal than other paradigms. To verify this, we conducted a critical experiment comparing Med-REFL against two strong alternatives: a **statistical sampling** approach using monte carlo tree search (MCTS) style rollouts, and a powerful **LLM-as-a-judge** using GPT-4.1. To ensure a fair comparison, all methods operated on the exact same set of candidate reasoning steps generated during our initial ToT exploration; the only variable was the mechanism act as 'action value' used to select the 'chosen' vs. 'rejected' paths for DPO. The results, presented in Table 7, are unequivocal. Med-REFL outperforms both the MCTS approach (even with a high number of simulations, 'k=20') and the strong GPT-4.1-as-judge baseline. Notably, while increasing the number of rollouts for

Table 7: Ablation study comparing Med-REFL against alternative data generation strategies on MedQA-USMLE. Med-REFL's deterministic, action-focused evaluation provides a superior learning signal.

| Method | Config. | ACC (%) |
|---|---|---|
| Base Model | - | 59.92 |
| | $k = 5$ | 60.76 |
| MCTS-Rollouts | $k = 10$ | 61.14 |
| | $k = 20$ | 61.21 |
| LLM-as-Judge | GPT-4.1 | 60.59 |
| **Ours** | - | **61.92** |

MCTS yields diminishing improvements, it never matches the performance of our method. This empirically validates our core hypothesis: the stable, holistic quality signal derived from our deterministic structural assessment is fundamentally more effective for teaching fine-grained reflection than either noisy statistical estimations or heuristic-based judgments. Detailed settings can be found in Appendix F.8.

## 4.6 COMPREHENSIVE ABLATION AND ROBUSTNESS ANALYSIS

Due to space constraints, we present extensive additional validations in the Appendix. Regarding data construction, we verify the superiority of ToT-based exploration over random rollouts (Appendix F.1) and analyze the impact of the generator model's scale (Appendix F.6). For training dynamics, we identify the optimal ratio of reasoning to reflection data (Appendix F.2) and demonstrate the necessity of contrastive learning by comparing DPO against SFT (Appendix F.4). Furthermore, we provide detailed performance analyses of the scoring model (Appendix F.3) and examine system robustness through hyperparameter sensitivity tests based on reasoning boundaries (Appendix F.9).

## 4.7 QUALITATIVE ANALYSIS

To complement our quantitative results, Figure 1 offers a qualitative snapshot of Med-REFL's impact on the model's internal reasoning process. More detailed case studies and analysis are provided in Appendix G for a deeper intuitive understanding.

## 5 DISCUSSION

In this section, we discuss and analyze the intriguing phenomena observed during the experiment.

**Domain Sensitivity**    When transfering Med-REFL to the logic puzzle domain, we observed that the hyperparameters effective for medical reasoning were insufficient for logic puzzles. Specifically, the K&K benchmark required a larger exploration budget (average step count of 6, median 7) compared to the medical domain (average step count 3.63) to generate high-quality data. We argue that this difference stems from the distinct nature of the tasks. The medical domain is primarily knowledge-intensive: the difficulty lies in retrieving and associating separate clinical entities. Once the correct key entity is identified, the deductive chain is often relatively short. In contrast, logic puzzles are reasoning-intensive, requiring deep, multi-step deductive chains where knowledge retrieval plays a minor role. This suggests that the exploration budget varies significantly by domain. Consequently, Med-REFL demonstrates the sensitivity of the parameters relative to the type of domain.

**Reflection as Attention Redirection**    We find that effective reflection in the medical domain often functions as a mechanism of attention redirection(e.g., Case Study G.2). In many error cases, the base model does not lack the necessary medical knowledge but fails to assign appropriate weight to contradictory evidence (e.g., overlooking a high MCV value in favor of a salient symptom like hematochezia). Med-REFL trains the model to revisit the context and specifically seek out counter-evidence that conflicts with its initial hypothesis. By re-weighting these overlooked cues, the model resolves clinical contradictions. This mechanism may explain why Med-REFL delivers the most substantial gains on complex multi-hop reasoning benchmarks (e.g., MedQA, MedXpert) compared to pure retrieval tasks (e.g., PubMedQA), as the former requires the holistic integration of conflicting information rather than simple fact lookup.

**Mitigating Fake Reflection via Contrastive Learning**    Another contribution of Med-REFL is demonstrating a scalable solution to the 'fake reflection' problem. We argue that Med-REFL effectively mitigates this issue through the specific design of our preference data construction and the DPO training objective. Our framework constructs two distinct types of reflective supervision: mid-reasoning pairs, which focus on micro-level step corrections, and post-reasoning pairs, which target macro-level path redirection. Mitigation of fake reflection is largely attributable to the generalization of this post-reasoning redirection capability. In constructing post-reasoning pairs, the 'Chosen' trajectory represents a valid transition from error to correctness, while the 'Rejected' trajectory contains the model's natural inertia to persist in error. Unlike SFT, teaching the model to mimic the linguistic style of reflection, DPO introduces a negative constraint. It explicitly penalizes the model's propensity to reinforce its initial flawed intuition. Consequently, the model learns to inhibit its confirmatory inertia and actively explores alternative logical paths when triggered to reflect. Furthermore, the Llama3.1's (+Med-REFL) performance gains on the OoD K&K puzzle suggests that the learned self-correction capability is not merely memorized domain knowledge. Instead, it represents a transferable meta-cognitive skill that generalizes across different contexts.

## 6 CONCLUSION

In this work, we introduced Med-REFL, a novel framework designed to tackle the prohibitive cost of verifying intermediate reasoning **within the complex domain of medicine**. It leverages a deterministic, structure-based method to automatically construct targeted preference data, effectively enhancing a model's self-correction capabilities. Our experiments demonstrate that Med-REFL achieves SOTA performance on the MedQA-USMLE benchmark and exhibits robust generalization across diverse medical datasets, with ablation studies confirming that it instills a genuine internalized self-correction capability. Furthermore, we believe that the core principles of Med-REFL are applicable to other domains facing similar verification bottlenecks, such as legal reasoning, and exploring this transferability will be a key direction for our future work. Our findings show that explicitly training for a high-quality internalized reflection capability is a crucial and effective pathway toward developing more reliable and trustworthy LRMs for high-stakes medical applications.

## ETHIC STATEMENT

Although Med-REFL demonstrates significant improvements in medical reasoning and reflection capabilities of Large Reasoning Models, the models enhanced by this method may still produce content that includes inaccuracies, biases, or incomplete reasoning, including potentially flawed intermediate reflection steps. Given the high-stakes nature of the medical domain, where errors can have severe consequences, the current models fine-tuned with Med-REFL are intended for research purposes and are not suitable for direct real-world clinical applications, diagnosis, or treatment decisions without rigorous validation and oversight by qualified human medical experts.

We impose strict limitations on the use of our Med-REFL framework and the models trained using it. They are not permitted for deployment in any critical care or diagnostic systems where inaccuracies could lead to patient harm or misinformed medical actions. We emphasize the ethical responsibility of any users or subsequent developers to adhere to these restrictions to safeguard patient safety and the integrity of medical practice.

## REPRODUCIBILITY STATEMENT

We are committed to ensuring the reproducibility of our work. To this end, we have made our code, the complete 33k preference dataset used for DPO training, and the fine-tuned model weights publicly available in an anonymous repository, as referenced in Appendix C. The core methodology of the Med-REFL framework is detailed in Section 2, with specific stages for trajectory generation and preference pair construction described in Subsections 2.1, 2.2, and 2.3. A comprehensive description of our automated data generation pipeline is provided in Appendix E.2, with specifics on the Tree-of-Thought exploration strategy in Appendix E.1. Furthermore, all prompts utilized for data generation and model interaction are fully documented in Appendix H. For experimental validation, Section 3 outlines the overall setup, while Appendix E.3 contains the specific hyperparameters for model fine-tuning (e.g., LoRA configurations, learning rate, batch size) and evaluation. The detailed configurations for our ablation studies (Section 4), including the comparative analysis against alternative strategies (Appendix F.8), are specified to allow for thorough verification of our claims.

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

# Table of Contents

## A   LLM Usage

LLMs were utilized as assistive tools during the preparation of this manuscript and the development of the accompanying code. Specifically, an LLM was employed for tasks related to writing and polishing the manuscript, including improving sentence structure, checking for grammatical errors, and enhancing the overall readability and clarity of the text. Additionally, an LLM was used in the code review process to help identify potential bugs, suggest structural improvements, and ensure code clarity and consistency.

It is important to emphasize that the LLM was not involved in the core research aspects of this work. All conceptual ideation, the design of the Med-REFL framework, the experimental methodology, and the analysis and interpretation of the results were conducted exclusively by the authors. The LLM's contribution was confined to improving the linguistic quality of the manuscript and the technical quality of the code, without any influence on the scientific content or outcomes.

The authors take full responsibility for all content presented in this paper, including any text or code assisted by the LLM. We have carefully reviewed all LLM-generated suggestions to ensure they are accurate and appropriately integrated, and to verify that the final work adheres to all ethical guidelines and is free from plagiarism or scientific misconduct.

## B   Related Works

### B.1   Outcome-Oriented Reasoning Enhancement

The most established paradigm for enhancing large language model (LLM) reasoning involves supervision based on final outcomes. This is primarily achieved through two training avenues: SFT and RL. In the medical domain, SFT is widely used to train models on high-quality chain-of-thought (CoT) data distilled from powerful teacher models, as seen in works like Huatuo-o1 and Reason-Med (Sun et al., 2025). Concurrently, outcome-based RL method such as GRPO (Shao et al., 2024) and DAPO (Yu et al., 2025) refines model policies by rewarding the final correct answer, a technique successfully transferred to medicine by Med-RLVR (Zhang et al., 2025a) and AlphaMed (Liu et al., 2025), and foundational to general reasoners like DeepSeek-R1 (DeepSeek-AI et al., 2025). While effective at improving task-specific performance, these outcome-oriented methods share a critical vulnerability: the learning signal is sparse and lacks process-level granularity (Zhang & Parkes, 2023). This makes them susceptible to 'reward hacking', where flawed logic that coincidentally produces a correct answer is positively reinforced (Lightman et al., 2023). Furthermore, the reliance on powerful teacher models for high-quality data distillation is often prohibitively expensive (Dasgupta et al., 2023). This paradigm, therefore, struggles to guarantee the learning of robust and sound reasoning processes.

### B.2   Process-Oriented Reasoning Enhancement

To address the lack of intermediate supervision, PRMs have been proposed to evaluate each step in a reasoning chain. This approach has been adapted for medical reasoning by training PRMs on stepwise quality labels, as demonstrated in Med-PRM (Yun et al., 2025) and MedS³ (Jiang et al., 2025). These labels are often generated via complex methods, such as statistical approximation from massive random rollouts (Askell et al., 2021; Jiang et al., 2025) or judgments from a complex retrieval-augmented generation (RAG) system (Yun et al., 2025). In some cases, as with MedReason (Wu et al., 2025), the reasoning process is constrained by a knowledge graph to ensure factual grounding. However, this granularity comes at a significant cost. The PRM approach typically requires training a separate, powerful reward model, introducing considerable computational overhead and architectural complexity (Yuan et al., 2024). More fundamentally, these frameworks are often optimized to identify the most correct reasoning path. This objective is insufficient for teaching a model how to reflect and recover from its own diverse errors (Huang et al., 2024a), hindering its ability to develop a robust, generalizable self-correction capability (Lan et al., 2024).

### B.3 REASONING PATH EXPLORATION AND EVALUATION

A third line of work leverages tree-search algorithms like monte carlo tree search (MCTS) (Kocsis & Szepesvári, 2006) and ToT to explore a vast space of reasoning possibilities as training data. This approach is exemplified by methods such as rStar-math (Guan et al., 2025), ASTRO (Kim et al., 2025), and the O1-Journey (Qin et al., 2024; Huang et al., 2024b) series. These methods are powerful for generating diverse reasoning data, including paths that contain trial-and-error, which is crucial for learning complex problem-solving. The primary limitation of many such approaches lies in the evaluation of the generated steps. Node values are often approximated via noisy statistical estimations from random rollouts (Hao et al., 2023), and the exploration can lead to "ineffective reflections"—jumps from one flawed reasoning state to another, without a guaranteed improvement in reasoning quality (Huang et al., 2024a). While these methods explore extensively, they often lack a robust mechanism to assess the value of a corrective action itself based on global information (Hao et al., 2023). In contrast, Med-REFL's deterministic, structure-based assessment is specifically designed to quantify the value of these corrective actions, ensuring that the learned reflection is consistently effective.

### B.4 SELF-CORRECTION AND CONFIRMATORY BIAS IN LRMS

Recent studies highlight a critical limitation in the reasoning capabilities of LRMs: the inability to genuinely self-correct due to deep-seated confirmatory bias. Kang et al. (2025) empirically demonstrate that model-generated reflections are predominantly confirmatory ('fake reflection'), with the correctness of the final output largely dictated by the initial reasoning attempt rather than subsequent reflection. This phenomenon is underpinned by 'reasoning inertia', where models develop an internal bias immediately upon processing the input, leading to 'overthinking' and redundant steps that reinforce the initial error rather than correcting it Dang et al. (2025). Consequently, without explicit supervision on *how* to correct, the reflection process frequently devolves into a 'post-hoc rationalization' of the initial failure Zhang et al. (2025b). Unlike previous approaches that rely on stochastic sampling or weak critique, Med-REFL explicitly constructs preference pairs to distinguish 'effective correction' from 'invalid reflection', directly supervising the model to break this confirmatory loop.

## C METHOD REPRODUCIBILITY STATEMENT

The code, data, and weights have been open-sourced and are available for reproduction.
Readers can visit the anonymous repository at:
https://anonymous.4open.science/r/Med-REFL-B5F8/
All the information has been anonymized, and the complete Med-REFL training dataset can be found in the repository's 'Data' folder.

## D LIMITATIONS AND FUTURE WORK

### D.1 LIMITATIONS

While Med-REFL demonstrates a significant step forward, we acknowledge several limitations that offer clear avenues for future research.

First, regarding the scope of our empirical validation, while our framework is designed to address a general class of problems characterized by a verification bottleneck, we acknowledge that the validation in this work is exclusively focused on the medical domain. We chose medicine due to its unique status as a 'crucible' for reasoning—its extreme complexity and high stakes provide the most rigorous possible test for our method. However, future work should explicitly validate the framework's transferability to other knowledge-intensive domains, such as legal case analysis or scientific literature review, to fully map its scope of applicability.

Second, the data generation process itself has inherent constraints. The ToT-based exploration is computationally more intensive than simpler RL paradigms. This cost necessitated that our preference dataset was primarily derived from a single, albeit large, benchmark (MedQA-USMLE). Consequently, the full scaling properties of Med-REFL—how its efficacy evolves with substantially

larger and more varied data sources—remain an open question. Furthermore, the effectiveness of our data generation is contingent on the base model's ability to produce a rich diversity of both correct and flawed reasoning paths; for problems where a model is either near-perfect or consistently fails, generating high-quality contrastive pairs can be challenging.

Finally, the scope of the 'reflection' we teach and evaluate is focused on logical self-correction. More complex facets of clinical reflection, such as integrating ethical considerations, patient preferences, or navigating profound uncertainty, are not addressed by our current framework. Similarly, our validation has predominantly been on multiple-choice question-answering formats. Future research should explore Med-REFL's efficacy on other task formats, such as open-ended diagnostic report generation or interactive clinical dialogues, to further probe the limits of the learned capabilities.

### D.2 FUTURE WORK

Our work on Med-REFL opens several exciting avenues for future research. A primary direction is to empirically validate the framework's generalizability, as we hypothesized, by applying it to other high-stakes domains with significant verification bottlenecks, such as legal case analysis and scientific literature review. Furthermore, we plan to extend our evaluation beyond multiple-choice formats to more complex, generative tasks, including open-ended diagnostic report generation and interactive clinical dialogues, to more rigorously assess the robustness of the learned reflection capabilities.

Another promising direction involves a large-scale study of Med-REFL's scaling properties to understand how performance evolves with an order-of-magnitude increase in preference data. This could be combined with research into making the learned reflection not only more effective but also more concise and computationally efficient. Finally, we believe significant potential lies in synergizing Med-REFL's reasoning engine with knowledge-injection methods like RAG, and in expanding the scope of 'reflection' itself to encompass more complex facets like ethical considerations and uncertainty management. Exploring these directions represents a key step toward creating more comprehensive and trustworthy AI reasoners.

## E SUPPLEMENTARY METHODS AND EXPERIMENTS

### E.1 TREE-OF-THOUGHT PATH SEARCHING FOR MEDICAL REASONING

This section details the tree-of-thought (ToT) based path searching mechanism employed to generate diverse reasoning trajectories for complex medical problems. This process of structured exploration forms the foundational stage for subsequent fine-grained evaluation and preference data construction within our Med-REFL framework. The ToT approach decomposes a given medical problem into a sequence of discrete reasoning steps or intermediate thoughts, which collectively form a tree-like structure. In this structure, each node represents an intermediate reasoning state, and a path from the root to any leaf node constitutes a complete reasoning trajectory towards a potential solution.

Formally, the generation of such a reasoning trajectory is an iterative process. Let $Q$ represent the initial medical problem description. The LLM, denoted as $\pi$, serves as the core reasoning engine. The process initiates by generating an initial reasoning step (or thought), $s_0$, based on $Q$:

$$s_0 \leftarrow \text{ProposeInitialStep}(Q, \pi) \tag{8}$$

Subsequent reasoning steps are generated iteratively, conditioned on the original problem $Q$ and the history of preceding steps. If $H_t = (s_0, s_1, \ldots, s_{t-1})$ denotes the sequence of thoughts generated up to step $t-1$, the next thought $s_t$ is proposed as:

$$s_t \leftarrow \text{ProposeNextStep}(Q, H_t, \pi) \tag{9}$$

This formulation ensures the LLM $\pi$ considers both the original problem and the accumulated reasoning history to maintain context and coherence. The process continues until a candidate solution is formed or a predefined maximum length $k_{max}$ is met. A complete reasoning trajectory leading to a candidate answer $a_Q$ is thus represented by the sequence $S_{final} = (s_0, s_1, \ldots, s_F)$, where $F < k_{max}$.

The prompts used for constructing ToT paths can be found in Appendix H.1.

---

**Algorithm 1** ToT-based Reasoning Trajectory Generation

---

**Input:** $Q$: Medical problem description; $k_{max}$: Maximum length of reasoning trajectory; $\pi$: Reasoning LLM

**Output:** $S_{solution}$: A solution trajectory, or **Failure**

1: $S_{current} \leftarrow ()$                         $\triangleright$ Initialize the current reasoning trajectory (sequence of steps)
2: $s_0 \leftarrow \text{ProposeInitialStep}(Q, \pi)$
3: **if** $s_0 = \text{null}$ **then**                              $\triangleright$ Cannot generate initial step
4:      **return Failure to Start**
5: **end if**
6: $S_{current} \leftarrow S_{current} \oplus s_0$                  $\triangleright$ Append $s_0$ to trajectory; $\oplus$ denotes concatenation
7: $i \leftarrow 0$                               $\triangleright$ Current depth or number of steps after initial
8: **while not** $W_{solve}(Q, S_{current})$ **and** $i < k_{max}$ **do**      $\triangleright$ $W_{solve}$ checks if $S_{current}$ is a complete solution
9:      $C_{next\_steps} \leftarrow \text{ProposeCandidateNextSteps}(Q, S_{current}, \pi)$      $\triangleright$ Generate set of candidate next steps
10:      **if** $C_{next\_steps} = \emptyset$ **then**
11:          **break**                           $\triangleright$ No further steps can be proposed from $S_{current}$
12:      **end if**
13:      $s_{best\_next} \leftarrow \text{SelectBestStep}(C_{next\_steps}, Q, S_{current}, W_{eval})$      $\triangleright$ $W_{eval}$ evaluates promise of candidates
14:      **if** $s_{best\_next} = \text{null}$ **then**
15:          **break**                           $\triangleright$ No promising step found among candidates
16:      **end if**
17:      $S_{current} \leftarrow S_{current} \oplus s_{best\_next}$
18:      $i \leftarrow i + 1$
19: **end while**
20: **if** $W_{solve}(Q, S_{current})$ **then**
21:      **return** $S_{current}$ as $S_{solution}$
22: **else**
23:      **return Failure to Solve** (e.g., $k_{max}$ exceeded or no valid path found)
24: **end if**

---

## E.2 DETAILED DESCRIPTION OF DATA CONSTRUCTION

Our preference dataset was constructed exclusively from the official MedQA-USMLE training set. The entire process was driven by a two-stage automated pipeline, which is detailed below. The official test split of the benchmark was reserved strictly for final performance evaluation.

**Stage 1: Seed Trajectory Generation and Scoring Model Training.** The initial step focused on generating a diverse set of reasoning paths.

- We utilized `Llama3.1-8B` with a Tree-of-Thoughts (ToT) exploration strategy to process the MedQA training set. This generated an initial pool of 45,000 reasoning trajectories.

- From this pool, we filtered down to 28,000 valid trajectories that contained a mix of both correct and incorrect reasoning paths.

- Each of these 28,000 trajectories was assigned a score based on its solution value, creating a dataset we denote as $D_{\text{SFT}}$. This dataset was subsequently used to train the scoring model introduced in Section 2.3.

**Stage 2: High-Quality Preference Pair Construction.** In the second stage, the 45,000 initial trajectories were first reconstructed into 10,000 distinct reasoning trees. These trees then served as the foundation for constructing our final DPO preference pairs using a more powerful model, `Qwen2.5-72B-Int4`. The dataset was partitioned by question, with approximately one-third of the questions allocated for 'Reasoning Enhancement' data and the remaining two-thirds for 'Reflection Learning' data. The reasons for this classification can be found in Appendix F.2

- **Reasoning Enhancement Pairs (12k):** These pairs were designed to teach the model general reasoning proficiency by contrasting high-quality correct reasoning paths against plausible but flawed ones.

- **Reflection Learning Pairs (21k):** These pairs specifically train the model's self-correction capabilities. They were generated by identifying an error in a reasoning path and contrast-

ing an *ineffective reflection* (which fails to correct the error) with an *effective reflection* (which successfully identifies and corrects the error). The generation logic for these pairs was as follows:

- If the *Error Locator* identified an incorrect step at a non-leaf node of the reasoning tree, a **mid-reasoning** reflection pair was generated.
- If the identified error was at a leaf node, we generated both a **mid-reasoning** pair (correcting the flawed leaf node to a correct one) and a **post-reasoning** pair (reflecting on the entire flawed trajectory from the error onward).

All subsequent modules, including the Error Locator and Reflector, were executed by the `Qwen2.5-72B-Int4` model via the vLLM (Kwon et al., 2023) engine. Detailed hyperparameters for this process are available in Appendix E.3.

### E.3 SUPPLEMENTARY EXPERIMENTS SETTINGS

**Key hyperparameters** Key hyperparameters for our value functions were set as follows: for the action value ($v_{act}$), the weights were $\lambda_1 = 0.4, \lambda_2 = 0.2, \lambda_3 = 0.4$;

**Evaluation and Training Details** For inference, we used a maximum generation length of 8192 tokens and randomized key sampling parameters, with temperature and ($top_p$) each drawn from the range [0.2, 1.0], to assess robust performance across various generation conditions. For efficient fine-tuning, all baseline models were adapted using LoRA (Hu et al., 2022) with a rank $r = 64$ and alpha $\alpha = 16$. Models were trained for 1 epoch with a learning rate of $1 \times 10^{-5}$ and a global batch size of 128. We utilized the LlamaFactory (Zheng et al., 2024) framework for training, which was optimized using DeepSpeed-ZeRO (Rajbhandari et al., 2020) Stage 3. All fine-tuning experiments were successfully conducted on a standard setup of 2 NVIDIA A100 (80GB) GPUs.

### E.4 DETAIL RESULTS FOR FIGURE 1 C) & FIGURE 1 D)

This section provides the detailed numerical data and corresponding analysis for the trends and competitive performance illustrated in Figure 1 c) and Figure 1 d) of the main paper.

Table 8: Comparison of model performance on MedQA, showing average token cost per question and accuracy before and after applying our method.

| Model | Avg. Token Cost | | Accuracy (%) | |
|---|---|---|---|---|
| | Original | w/ our | Original | w/ our |
| Huatuo-o1 | 657.30 | 857.52 | 69.59 | 73.72 |
| Llama3.1-8B | 250.46 | 540.73 | 59.92 | 65.74 |
| MedReason | 761.35 | 885.89 | 66.27 | 70.16 |
| AlphaMed | 339.74 | 384.95 | 67.79 | 69.17 |
| Qwen2.5-7B | 252.14 | 239.55 | 57.11 | 59.70 |
| Deepseek-Distill-8B | 1043.82 | 1530.12 | 48.85 | 55.00 |
| Ultra-Medical | 517.79 | 618.47 | 71.34 | 73.08 |

**Analysis of Reasoning Depth (for Figure 1 c))** As discussed in the main text, our Med-REFL framework is designed to enhance the depth and quality of the model's reasoning process. Table 8 presents the detailed data underlying Figure 1 c), quantifying the relationship between the average token cost per question and the final inference accuracy on the MedQA benchmark. The results strongly support our hypothesis that improved performance is structurally linked to more elaborate reasoning.

For the vast majority of models tested, the application of Med-REFL (denoted as 'w/ our') resulted in a concurrent increase in both average token cost and accuracy. For instance, the reason-heavy

model **Huatuo-o1** saw its token cost increase from 657.30 to 857.52, with its accuracy rising significantly from 69.59% to 73.72%. Similarly, **Llama3.1-8B**, a general-purpose model, experienced a substantial accuracy gain of +5.82% (from 59.92% to 65.74%) as its reasoning traces became more detailed (token cost increased from 250.46 to 540.73). This pattern holds for other specialized models like **MedReason**, **AlphaMed**, and **Ultra-Medical**. The most dramatic increase is seen in **Deepseek-Distill-8B**, where a notable +6.15% accuracy improvement was accompanied by an increase in token cost, indicating a profound enhancement of its reasoning process.

An interesting exception is **Qwen2.5-7B**, where the token cost slightly decreased after applying our method, yet accuracy still improved by +2.59%. This suggests that for certain models, Med-REFL can also improve reasoning *efficiency*, enabling them to reach a correct conclusion more directly. Overall, the data in Table 8 validates that the performance gains from Med-REFL stem from fostering a more detailed, in-depth, and effective reasoning process.

Table 9: Performance comparison of Med-REFL-8B against other models. Our model (**Med-REFL-8B**) is placed at the bottom of the 8B and 10-20B categories for direct comparison, separated by a dashed line. The best score within each category (8B models and 10-20B models) is bolded.

| Model | MedQA (5op) | MedMCQA | PubMedQA | GPQA (M) | MMLU-Pro (M) | MedXpert-U | MedXpert-R |
|---|---|---|---|---|---|---|---|
| *Reference Model (>20B)* | | | | | | | |
| Qwen-3-30B-a3B | 81.74 | 69.83 | 74.20 | 52.31 | 77.95 | 21.73 | 19.88 |
| *10B-20B Models* | | | | | | | |
| Qwen-3-14B | **79.03** | **68.26** | 73.90 | 54.62 | **72.81** | 19.86 | **20.20** |
| Microsoft-Phi-3-14B | 69.60 | 62.92 | 69.10 | 52.05 | 69.77 | 17.66 | 14.08 |
| Google-Gemma-3-12B | 65.67 | 58.00 | 60.60 | 51.03 | 68.14 | 16.81 | 14.24 |
| Qwen-2.5-14B | 65.12 | 61.55 | 73.43 | 53.50 | 59.64 | 13.92 | 12.95 |
| Mistral-Nemo-14B | 47.45 | 48.72 | 63.35 | 39.74 | 52.08 | 13.75 | 12.01 |
| **Med-REFL-8B** *(for comp.)* | 73.72 | 64.66 | **81.32** | **56.80** | 64.97 | **20.02** | 17.78 |
| *General-purpose 8B Models* | | | | | | | |
| Qwen-3-8B | 73.37 | 63.97 | 72.10 | 55.04 | *75.83* | 19.19 | 16.28 |
| Llama-3.1-8B | 59.92 | 57.61 | 76.26 | 45.16 | 57.56 | 16.11 | 14.14 |
| Qwen-2.5-8B | 57.11 | 54.52 | 74.12 | 43.34 | 61.36 | 10.79 | 12.45 |
| Mistral-8B | 48.00 | 49.13 | 64.65 | 33.33 | 53.16 | 14.52 | 12.17 |
| *Specially-trained 8B Models* | | | | | | | |
| UltraMedical-3.1-8B | 71.34 | 63.30 | 78.60 | 45.76 | 63.06 | 15.68 | 15.87 |
| Huatuo-1-8B | 69.59 | 62.13 | 79.02 | 50.67 | 61.87 | 15.31 | 16.85 |
| AlphaMed-8B | 67.79 | 61.22 | 78.03 | 53.40 | 63.29 | 20.24 | 21.91 |
| MedReason-8B | 66.27 | 58.98 | 77.17 | 45.64 | 59.14 | **22.55** | **22.31** |
| Deepseek-Distill-8B | 48.85 | 49.51 | 69.30 | **62.43** | 53.83 | 13.07 | 11.98 |
| **Med-REFL-8B** | **73.72** | **64.66** | **81.32** | 56.80 | **64.97** | 20.02 | 17.78 |

**Competitive Performance Analysis (for Figure 1 d))** Figure 1 d) highlights the exceptional competitive standing of our **Med-REFL-8b** model (Huatuo-o1 trained with Med-REFL). Table 9 provides the comprehensive, head-to-head numerical results that form the basis of this analysis, comparing our model against both same-class and larger-scale models across a wide array of medical benchmarks.

Within the 8B parameter class, **Med-REFL-8b** establishes itself as a state-of-the-art model. As shown in the table, it achieves the highest scores among all general-purpose and specially-trained 8B models on several of the most challenging benchmarks, including **MedQA (73.72%)**, **MedMCQA (64.66%)**, **PubMedQA (81.32%)** and **MMLU-Pro (64.97%)** (Qwen3-8B has an abnormal value for this benchmark). This dominant performance underscores the effectiveness of our framework in pushing a strong base model (Huatuo-o1) to new performance heights.

Furthermore, the results demonstrate that **Med-REFL-8b** is not only a leader in its own weight class but is also highly competitive with, and in many cases superior to, models in the much larger 10-20B parameter range. For example, its performance on PubMedQA (81.32%) surpasses all listed 10-20B models, including the powerful **Qwen-3-14B** (73.90%). On the complex reasoning benchmark GPQA, its score of 56.80 is also higher than all models in the 10-20B category. This evidence strongly supports the conclusion from our main analysis: Med-REFL provides a resource-efficient pathway to achieving top-tier medical reasoning capabilities, rivaling models with nearly double the parameters without the need for massive model scaling. The performance of **Qwen-3-30B-a3B** is

included as a high-level reference to situate our model's performance within the current landscape of leading-edge models.

# F  SUPPLEMENTARY ANALYSES AND ABLATION STUDIES

## F.1  ANALYSIS OF ToT-BASED VS. RANDOM ROLLOUT DATA GENERATION

To validate Med-REFL's ToT-based data construction—a strategy designed to generate higher-quality, nuanced training signals than conventional methods—we compared its DPO data against pairs derived from a traditional random rollout ($RO$) approach. For the $RO$ baseline, multiple reasoning paths per question were sampled and scored using our scoring model ($V_{CoT}$ from Section 2.3) to select preference pairs ($RO_{Llama}$: 14k, $RO_{Huatuo}$: 10k). As shown in Table 10, models trained with Med-REFL DPO data significantly outperform those trained with $RO$ data across both Huatuo-o1 (73.72% vs. 69.97%) and Llama3.1-8B (65.74% vs. 62.21%). This outcome highlights the superiority of Med-REFL's structured, ToT-based trajectory sampling and fine-grained evaluation. Such a principled approach is crucial for generating more effective preference pairs for complex reasoning, directly addressing the motivation to overcome limitations of less nuanced or purely result-oriented data generation strategies.

Table 10: Ablation study on MedQA-USMLE comparing Med-REFL (our ToT-based DPO data) against DPO data from Random Rollouts (RO). Results highlight the performance advantage of Med-REFL's structured trajectory sampling and fine-grained evaluation over the RO approach.

| Training Data | Huatuo-o1 | Llama3.1-8B |
|---|---|---|
| Base Model | 69.59 | 59.92 |
| Random Rollout (RO) | 69.97 | 62.21 |
| **Med-REFL (Ours)** | **73.72** | **65.74** |

## F.2  ANALYSIS OF THE DATA COMPONENT RATIO

To understand the interplay between general reasoning enhancement and targeted reflection training, we conducted an experiment to find the optimal mixing ratio of our two data types. We trained Llama3.1-8B on the MedQA-USMLE development set using DPO data mixed at various ratios of 'Reasoning Enhancement' pairs to 'Reflection Enhancement' pairs. As shown in Table 11, the model's performance peaked at a 1:2 ratio. This finding suggests that while general reasoning data is an important foundation, a heavier emphasis on our novel reflection-specific training data is key to maximizing performance gains. This result justifies our final dataset composition of splitting the seed questions for data generation at a 1:2 ratio.

Table 11: Impact of the ratio of Reasoning-to-Reflection data on Llama3.1-8B's performance on the MedQA-USMLE development set. A 1:2 ratio yields the best performance.

| Metric | 1:1 | 1:2 | 1:3 | 1:4 |
|---|---|---|---|---|
| Accuracy (%) | 61.92 | **62.43** | 62.11 | 62.21 |

## F.3  ANALYSIS OF THE SCORING MODEL FOR TRAJECTORY RANKING

To validate the efficacy of our trajectory evaluation metric ($v_{sol}$) and justify the necessity of the scoring model introduced in Section 2.3, we conducted an experiment to analyze its ability to filter high-quality reasoning paths. We trained scoring models of varying sizes (0.5B, 1B, and 3B) on the dataset $D_{SFT}$ (Section 2.3). These models were then tasked with scoring and ranking all trajectories generated by the ToT process, after which a multi-vote strategy was applied to the top-k selected paths to determine the final answer. This approach was benchmarked against a baseline that applies multi-vote across all generated trajectories without any filtering. The results, presented in Figure 3, reveal two key insights. First, as shown in Figure 3a, the effectiveness of the scoring model is

contingent on its capacity. While smaller 0.5B and 1B models degraded performance compared to the 57.03% baseline, the 3B model improved accuracy to 59.63% (with k=5). This confirms that our $v_{sol}$ metric provides a meaningful learning signal, which requires a sufficiently powerful model to leverage effectively. Second, Figure 3b illustrates that for the 3B model, performance peaks at k=6 with an accuracy of 59.86%, demonstrating an optimal trade-off. An overly small k increases reliance on a single, potentially flawed top prediction, while a larger k dilutes the benefits of filtering by including lower-quality paths. This analysis validates our choice to employ a dedicated scoring model for robustly ranking reasoning trajectories.

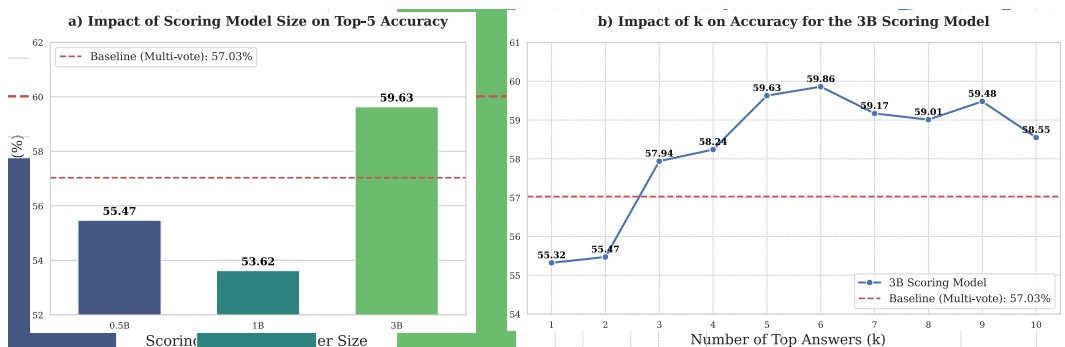

Figure 3: Analysis of the scoring model's effectiveness. a) Impact of scoring model parameter size (Qwen2.5-0.5B, Llama3.2-1B and Llama3.2-3B) on accuracy with a fixed k=5. b) Impact of the number of top answers (k) on the 3B scoring model's accuracy.

### F.4    ABLATION ON THE ROLE OF NEGATIVE SAMPLES VIA FINE-TUNING STRATEGY: DPO VS. SFT

We investigate which data components and learning mechanisms contribute most significantly to the performance gains. Our central hypothesis is that the key to enhancing reasoning lies not in merely imitating correct solutions, but in learning to distinguish correct reasoning from plausible but flawed alternatives. To test this, we conducted a crucial experiment comparing the effect of SFT against our full DPO approach. As shown in Table 12, fine-tuning the model via SFT on only the positive examples ('chosen' paths) yielded a slight improvement. In stark contrast, the full Med-REFL approach, which uses DPO to learn from the preference pairs of 'chosen' vs. 'rejected' paths, delivered the full, substantial performance improvement. This result offers compelling evidence for our core thesis. It demonstrates that the dnegative samples identified by our framework are not noise, but a vital learning signal. The significant performance uplift is unlocked only through contrastive learning via DPO, which explicitly teaches the model to recognize and move away from erroneous reasoning patterns. Simply memorizing correct paths is insufficient; the model must learn what not to do. Further analyses in Appendix F.2&F.5, we use more ablation experiments to explain why the 1:2 ratio of 'Reasoning' to 'Reflection' and their respective effects.

Table 12: Ablation study on the fine-tuning method. ACC for accuracy.

| Training Strategy on Llama3.1-8B | ACC (%) |
|---|---|
| Original (Base Model) | 59.92 |
| + SFT (on all 'chosen' paths) | 60.32 |
| **+ DPO (Full Med-REFL)** | **65.74** |

### F.5    ABLATION ON DATA COMPONENTS

To further dissect the contribution of our data components, we analyzed their independent impact by training models on only one type of data at a time. We fine-tuned both the general-purpose Llama3.1-8B and the domain-adapted Huatuo-o1 on 'Reasoning Enhancement' data only, and in a separate

run, 'Reflection Enhancement' data only. The results in Table 13 reveal a crucial, model-dependent effect. While both data components are independently valuable, the already strong, domain-adapted Huatuo-o1 benefits more from explicitly training its self-correction capability (+1.34% gain from Reflection data vs. +0.57% from Reasoning data). This powerfully validates our core hypothesis: for models with a strong baseline reasoning faculty, directly enhancing their reflective capabilities is a highly effective improvement strategy.

Table 13: Performance on the MedQA-USMLE development set when training with only one type of enhancement data. Note the significant impact of Reflection Enhancement on the domain-adapted Huatuo-o1 model. (Orig. means original accuracy. Only Refl. refers only to using reflection learning data.)

| Model | Original Accuracy | Only Reasoning | Only Reflection |
|---|---|---|---|
| Llama3.1-8B | 59.92% | **61.24%** | 60.85% |
| Huatuo-o1 | 69.59% | 70.16% | **70.93%** |

## F.6 ANALYSIS OF THE GENERATOR MODEL SCALE

Table 14: Ablation on the choice of model for initial ToT path generation, evaluated on a subset of the development set. The 8B model provided the optimal trade-off for generating a large, diverse dataset.

| Model | Generator Dev Acc. | Generated DPO Pairs | Acc. Gain on Llama3.1-8B |
|---|---|---|---|
| Llama3.1-70B | ∼80% | 1,576 | +0.86% |
| **Llama3.1-8B** | **∼63%** | **5,498** | **+1.73%** |
| Qwen2.5-3B | ∼36% | 2,174 | +0.45% |

A key choice in our pipeline is the use of a medium-sized model (Llama3.1-8B) for the initial ToT path exploration. To justify this, we performed an ablation study comparing the final performance gain when using DPO data generated from models of varying capabilities. As shown in Table 7, our choice of an 8B model occupies a crucial 'sweet spot'. The much larger 70B model, while powerful, was 'too strong' for this task; it rarely produced diverse, plausible errors, resulting in a small, imbalanced preference dataset and suboptimal training. Conversely, the smaller 3B model was 'too weak', struggling to generate high-quality correct paths. The 8B model provided the ideal balance, generating a rich and diverse set of both correct and incorrect paths, which is essential for high-quality DPO dataset construction.

## F.7 FURTHER ANALYSIS OF THE INTERNALIZED REFLECTION CAPABILITY

A notable pattern that emerged from our main results was that Med-REFL's performance gains were most pronounced on reason-heavy models like Huatuo-o1. We hypothesized that this is because such models, through prior domain-specific training, have already developed a rudimentary 'reflection' paradigm. This led us to a critical hypothesis, illustrated in Figure 4: the full benefit of our reflection-specific training is unlocked only when the model is explicitly prompted or trained to engage its reflection faculty. It suggests that the learned capability might be contingent on the inference strategy. To test this hypothesis, we designed a controlled experiment evaluating our fine-tuned Llama3.1-8B model under two distinct prompting conditions: a standard prompt instructing it to 'think step-by-step', and a second prompt that explicitly added a 'critical reflection' stage after the initial thinking. These prompts were applied to the base model, a model fine-tuned only on general reasoning data (Med-REFL w/o RC), and the model fine-tuned on our complete dataset including reflection-correction pairs (Med-REFL Full). The results, detailed in Table 15, are revealing. Under standard prompting, the full Med-REFL model already shows a clear advantage over the model trained without reflection-correction (RC) data (65.74% vs. 65.17%). However, when reflection

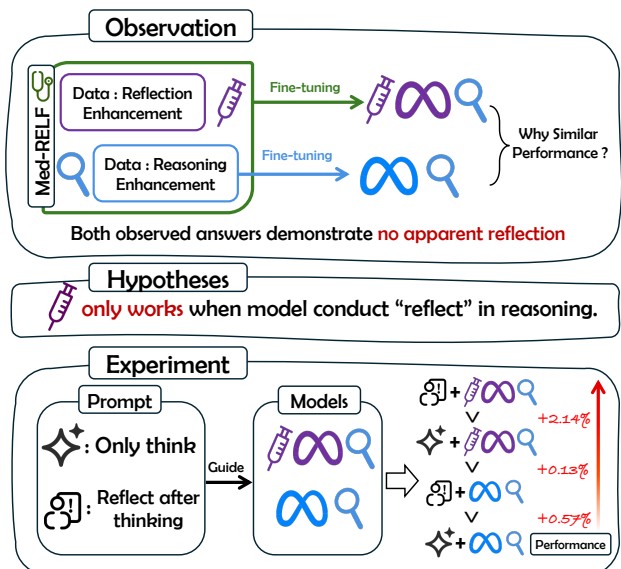

Figure 4: Experimental design for testing reflection capability under prompt guidance. This setup investigates how explicit prompting strategies ('Reflect after thinking' vs. 'Think w/o reflect') affect the performance of Llama3.1-8B fine-tuned with Med-REFL data that either includes ('Med-REFL (Full)', + ) or excludes ('Med-REFL (w/o RC)', ) dedicated reflection-correction pairs . The experiment aims to demonstrate that guided reflection is key to realizing the full benefits of reflection-specific training data.

is explicitly activated with the 'Reflect after think' prompt, the performance of the model trained on our full dataset surges by a significant +2.41% to 68.15%. In contrast, the base model and the model trained without our RC data saw much smaller gains from the same prompt. This outcome provides powerful evidence for our hypothesis. It confirms that Med-REFL does not merely teach models to mimic a reflection-like writing style; it instills a genuine, internalized cognitive capability for self-correction. This capability can remain latent under standard conditions but can be explicitly activated when required, demonstrating a deeper and more robust form of learning. This finding marks a crucial step towards creating AI systems that can be reliably guided to be more thoughtful and self-aware in their problem-solving processes.

Table 15: Ablation study on MedQA-USMLE illustrating the impact of including dedicated reflection-correction (RC) training data within Med-REFL. 'Med-REFL (w/o RC)' denotes training with DPO pairs focused on general reasoning discernment only, while 'Med-REFL (Full)' incorporates all data components, including those specifically targeting reflection and error correction.

| Training Data Variant | Huatuo-o1 | Llama3.1-8B |
|---|---|---|
| Base Model | 69.59 | 59.92 |
| Med-REFL (w/o RC) | 70.63 | 65.17 |
| **Med-REFL (Full)** | **73.72** | **65.74** |

## F.8 Detailed Experimental Setup for 'Comparison with Alternative Strategies' in Ablation Study

To ensure a fair and rigorous comparison between Med-REFL and alternative data generation paradigms, we designed a controlled experimental environment where all methods operated on a unified set of reasoning steps. This appendix details the setup for this critical ablation study.

**General Setup and Data Selection.** All fine-tuning experiments for this study were conducted on the **Llama3.1-8B** base model. The data source was a pool of 1614 instances of incorrect reasoning nodes, denoted as 'wrong nodes', sampled from the full set of error-containing paths generated during our initial tree-of-thoughts exploration on the MedQA-USMLE training set. To guarantee a fair comparison, all methods were tasked with evaluating the exact same candidate pool for each instance. This **unified candidate pool** consisted of the 'wrong nodes' itself plus all of its sibling nodes from the original ToT tree. The shared ancestor path leading to these nodes is denoted as 'history'.

**Method A: Med-REFL (Control Group).** Our control group follows the original Med-REFL methodology detailed in Section 3 of the main paper. For each candidate 'error' node in the unified pool, we deterministically calculated its $v_{action}$ score with 'next jump' nodes in other paths. The DPO preference pair was then constructed by contrasting the paths originating from the highest-scoring ('chosen') and lowest-scoring ('rejected') nodes.

**Method B: MCTS-Multiple Rollouts (Experimental Group).** This method was designed to be completely decoupled from the global structural information of the ToT. Same as method A, method B&C also include the 'reflection' in DPO pair construction.

- **Scoring:** For each candidate node, we performed $k$ independent, random rollouts, with $k \in \{5, 10, 20\}$. The score $p$ for each node was calculated as $p = (N_{correct} - N_{incorrect})/k$, where $N$ is the number of correct or incorrect rollouts.

- **DPO Pair Construction:** We identified the nodes with the highest and lowest scores, $p_{max}$ and $p_{min}$. The 'chosen' trajectory for the DPO pair was constructed from a randomly selected successful rollout initiated from the highest-scoring node. The 'rejected' trajectory was constructed from a randomly selected failed rollout initiated from the lowest-scoring node. If a selected node had no successful (or failed) rollouts, the DPO pair for that data point was discarded.

**Method C: LLM-as-Judge (Experimental Group).** This method relied on the external, outcome-agnostic judgment of GPT-4.1.

- **Scoring:** For each candidate node, we prompted GPT-4.1 to assign a quality score, $Score_{judge}$, on a scale of 1 to 5 based on its perceived medical accuracy and logical coherence, without any knowledge of the final answer.

- **DPO Pair Construction:** The nodes with the highest and lowest judge scores were identified. A new, complete reasoning path was then generated starting from each of these two nodes to form the final 'chosen' and 'rejected' trajectories for the DPO pair.

---

**Prompt for MCTS Rollouts (performing a single rollout)**

You are a medical expert completing a reasoning process for a USMLE-style question. You are given the original **QUESTION**, its **OPTIONS**, and a fixed reasoning **PATH** to follow.

---

**Your Task:**
Your task is to continue this reasoning **PATH** to its logical conclusion and select the single best answer from the **OPTIONS**.

**Output Format:**
Think step-by-step. At the very end of your entire response, you **MUST** state the final answer in the following exact format on a new line:
```
Final Answer: [OPTION]
```

---

**Inputs:**

```
QUESTION:
{question}

PATH:
{history}
{next_step}
```

---

**Prompt for LLM-as-Judge Step Evaluation**

You are an expert medical educator and a master of logical reasoning, tasked with evaluating a single step in a student's problem-solving process for a USMLE question.

---

**Instructions:**
Evaluate the **CURRENT_STEP** on a scale of 1 to 5 based on medical accuracy, logical coherence, and strategic value.

**Scoring Scale:**

- **1 (Flawed/Harmful):** Medically incorrect or logically fallacious.

- **2 (Weak):** Correct but unhelpful, irrelevant, or shows poor prioritization.

- **3 (Acceptable):** Logical and correct, but generic or suboptimal.

- **4 (Strong):** Logical, accurate, and strategically advances the reasoning.

- **5 (Excellent):** Strong, and also demonstrates exceptional insight.

**Output Format:**
Your output **must** be a single, valid JSON object with two keys: `reasoning` and `score`.

```
{
  "reasoning": "Your brief explanation here...",
  "score": <your integer score from 1 to 5>
}
```

## F.9 Sensitivity and Robustness Analysis: A Reasoning Boundary Perspective

In this section, we analyze the sensitivity of our framework to key hyperparameters, specifically sampling parameters (seeds, temperature), search budget (branching factor, depth), and the action value weights ($\lambda$). We propose to view these parameters through the lens of the model's **reasoning boundary** (Chen et al., 2024).

Since Med-REFL relies on learning the transition from erroneous states to correct states (i.e., self-correction), effective training data can only be generated from problems that lie within the model's

reasoning boundary: the problem must be difficult enough to induce errors, yet solvable enough for the model to eventually find at least one correct path via search. Hyperparameters essentially determine the extent of this boundary explored during data generation.

### F.9.1 STABILITY ACROSS SAMPLING PARAMETERS

The randomness in trajectory generation, controlled by random seeds and temperature ($T$), theoretically affects the specific paths explored in a single run. However, in the context of our ToT approach, this sensitivity is minimal. ToT functions similarly to a high $Pass@K$ sampling strategy, where multiple rollouts are aggregated.

As long as the sampling parameters allow for sufficient diversity (e.g., $T > 0$), the probability of discovering a correct path for a problem within the model's reasoning boundary remains stable. Therefore, variations in seeds or slight fluctuations in temperature primarily shift *which* valid path is found first, rather than *whether* a path is found, ensuring the robustness of our method against stochastic noise.

### F.9.2 IMPACT OF SEARCH BUDGET AND TRAJECTORY DISTRIBUTION

The search budget, defined by the branching factor ($B$) and maximum depth ($D$), directly constrains the reachable reasoning space. Setting these values too low artificially contracts the reasoning boundary, preventing the model from solving complex problems that require deeper or broader reasoning. Conversely, excessive values incur diminishing returns in computational efficiency.

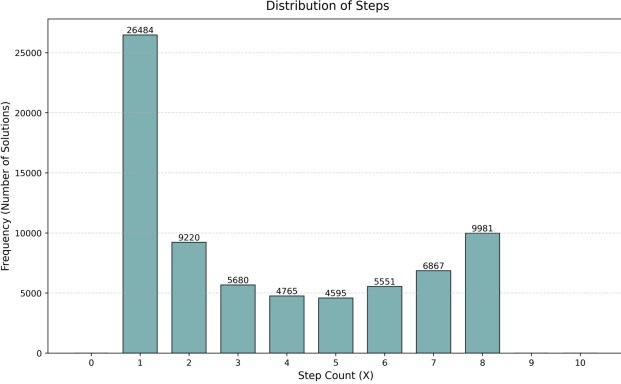

Figure 5: **Distribution of Reasoning Step Counts.** An analysis of valid solutions generated during the ToT phase reveals a long-tail distribution with a mean step count of 3.63 (Median=3, Std=2.66). This empirical evidence confirms that our maximum depth setting of $D = 8$ is sufficient to cover the vast majority of reasoning trajectories without truncating the model's capability.

We analyzed the distribution of reasoning steps for valid solutions generated under our settings ($B = 3, D = 8$). As shown in Figure 5, the distribution is long-tailed with a mean length of 3.63 steps and a median of 3. The cumulative coverage at 8 steps is near 100%. This confirms that our search budget is sufficient to capture the natural reasoning length of the model, effectively balancing exploration capability with computational overhead.

### F.9.3 SENSITIVITY OF ACTION VALUE WEIGHTS ($\lambda$)

The action value $v_{act}$, defined in Equation 4, guides the preference learning by weighting three components: step value ($\lambda_1$), solution value ($\lambda_2$), and remaining value ($\lambda_3$). We interpret our chosen ratio of $\lambda_1 : \lambda_2 : \lambda_3 = 2 : 1 : 2$ from both qualitative and quantitative perspectives.

**Qualitative Analysis.** In the context of mid-reasoning reflection, the model evaluates a transition between nodes rather than a finalized solution.

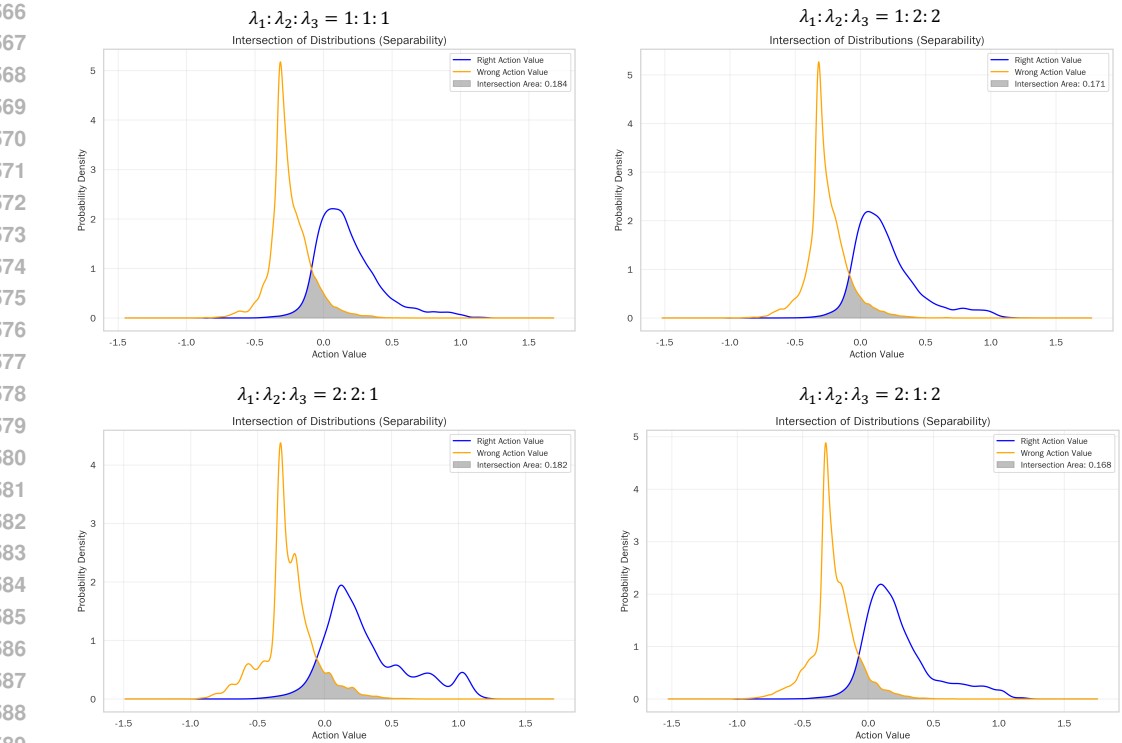

Figure 6: **Separability Analysis of Action Value Distributions under Different $\lambda$ Settings.** We plotted the probability density functions of action values for "Right Actions" (Blue) and "Wrong Actions" (Orange). The intersection area represents the confusion region. Our chosen setting of $\lambda_1 : \lambda_2 : \lambda_3 = 2 : 1 : 2$ achieves the minimum intersection area (0.168), indicating maximum separability and the strongest signal for distinguishing effective reflections.

- We assign higher weights to **Step Value ($\lambda_1$)** and **Remaining Value ($\lambda_3$)** because the primary goal of reflection is to immediately correct a local error (Step Value) and ensure the corrected path has high potential to lead to a solution (Remaining Value).

- **Solution Value ($\lambda_2$)** is weighted lower because it represents a global outcome that may be distant and less discriminative for intermediate corrective actions.

**Quantitative Separability Analysis.** To validate this empirically, we analyzed the discriminative power of different $\lambda$ configurations. We plotted the distribution of calculated action values for known "effective reflections" (right action) versus "ineffective reflections" (wrong action) and calculated the intersection area between the curves (Figure 6). A smaller intersection area implies better separability and a cleaner learning signal for the DPO process.

- $\lambda(1 : 1 : 1) \rightarrow$ Intersection Area: 0.184

- $\lambda(1 : 2 : 2) \rightarrow$ Intersection Area: 0.171

- $\lambda(2 : 2 : 1) \rightarrow$ Intersection Area: 0.182

- $\lambda(2 : 1 : 2) \rightarrow$ **Intersection Area:** 0.168 **(Optimal)**

The results demonstrate that the 2:1:2 setting maximizes the distinction between correct and incorrect reflective actions, thereby providing the most robust supervision signal.

# G   CASE STUDY

To provide a more intuitive and granular understanding of how the Med-REFL framework enhances the complex medical reasoning and self-correction capabilities of Large Language Models (LLMs), this appendix presents three detailed case studies. These cases are selected to respectively showcase:

- The qualitative shift in a model's reflective capability before and after Med-REFL fine-tuning when explicitly prompted;
- How a Direct Preference Optimization (DPO) pair teaches the model to prefer effective reflection over flawed reasoning;
- The profound evolution in the chain-of-thought of a state-of-the-art (SOTA) model, Huatuo-o1, after being fine-tuned with Med-REFL;

## G.1   THE INTERNALIZATION OF REFLECTION CAPABILITY VIA MED-REFL

This case study aims to validate that Med-REFL instills a genuine, activatable self-correction capability rather than merely teaching the model to mimic the superficial form of reflection. We compare the performance of the Llama3.1-8B model on the same question (Figure 7) before and after Med-REFL fine-tuning, specifically when prompted to engage in 'critical reflection'.

| Question(Q1) |
| --- |
| Question:
A 47-year-old executive schedules an appointment his physician for a routine medical check-up. He currently has no complaints and claims to be "as fit as a fiddle." The physical examination findings are unremarkable, except for a mid-systolic murmur heard in the 2nd left intercostal space that radiates to the carotids on auscultation. The physician instructs the patient to stand from a supine position with the stethoscope still placed on his chest. Which of the following changes would occur with this maneuver?
Options:
A. An increase in right atrial pressure
B. An increase in left ventricular end-diastolic pressure
C. An increase in pulmonary capillary wedge pressure
D. A reduction in the slope of the pacemaker potential
E. A reduction in diastolic filling time |

Figure 7: The clinical scenario for Question 1, involving hemodynamic changes upon standing.

**Before Med-REFL Fine-Tuning (Invalid Reflection)**   Prior to fine-tuning, when prompted to reflect, the model attempts to re-evaluate its logic but ultimately reinforces its initial flawed reasoning (see Figure 8).

- **Initial Thought Process:** The model correctly identifies that standing from a supine position reduces venous return due to gravity. However, it incorrectly focuses on the body's compensatory mechanisms, reasoning that increased sympathetic tone would elevate right atrial pressure to maintain cardiac output, thus favoring 'Option A'.
- **Critical Reflection (Ineffective):** During the reflection phase, the model acknowledges that standing reduces preload, which appears to contradict 'Option A'. However, it fails to recognize this as the dominant hemodynamic effect. Instead, it doubles down on its initial error with a flawed justification: 'sympathetic activation enhances venous return to the right atrium, potentially increasing...murmur intensity'. This reflective process does not correct the error but rather constructs a more complex defense for the wrong answer.
- **Final Conclusion:** The model incorrectly concludes with answer 'A'.

**After Med-REFL Fine-Tuning (Meaningful Reflection)**   After being fine-tuned with Med-REFL, the model demonstrates a genuine and effective self-correction capability, as shown in its reasoning process (Figure 9).

- **Initial Thought Process:** The model again analyzes the initial change of reduced venous return from standing. It considers the possibility of increased right atrial pressure due to compensation ('Option A') but also considers that reflex tachycardia could lead to a reduction in diastolic filling time ('Option E').
- **Critical Reflection (Effective):** In a pivotal moment of self-correction, the model explicitly identifies its initial line of thought as 'flawed'. It correctly recognizes that the **primary effect** of standing is the pooling of blood in the lower extremities, which directly leads to a

---

**Wrong Case in Internalization of Reflection Capability Ablation Study(Q1)**

## Thinking
I'm analyzing a 47-year-old male with a mid-systolic murmur in the 2nd left intercostal space, radiating to the carotids, suggesting a patent foramen ovale (PFO) or atrial septal defect (ASD). The maneuver involves standing from a supine position, and I need to identify the primary physiological change. Standing reduces venous return due to gravity, pooling blood in the lower extremities. The body compensates by increasing sympathetic tone, which may elevate right atrial pressure to maintain cardiac output (Option A). Alternatively, reduced venous return could decrease left ventricular end-diastolic pressure (Option B) or pulmonary capillary wedge pressure (Option C). Option D (pacemaker potential slope) seems unrelated, while Option E (reduced diastolic filling time) could occur due to increased heart rate. Now I lean toward Option A, as increased right atrial pressure aligns with compensatory mechanisms and may affect shunting in PFO/ASD.

## Critical Reflection
My initial reasoning assumed PFO/ASD causes the murmur, but the radiation to the carotids also suggests aortic stenosis, though the location favors PFO/ASD. Standing reduces preload, decreasing left-sided pressures (ruling out Options B and C). Option D is irrelevant, as pacemaker potential slope isn't directly affected by posture. Option E is plausible due to increased heart rate shortening diastole, but it's secondary to the primary hemodynamic change. Option A (increased right atrial pressure) is most consistent, as sympathetic activation enhances venous return to the right atrium, potentially increasing shunting and murmur intensity in PFO/ASD. No other conditions better explain the findings.

## Conclusion
The murmur suggests PFO or ASD. Standing reduces venous return, triggering sympathetic compensation that increases right atrial pressure to maintain cardiac output. This directly impacts the right-sided pressures relevant to the murmur's pathophysiology.

**Final Answer: A**

---

Figure 8: Invalid reflection chain for Q1 from the baseline Llama3.1-8B model. Despite attempting to reflect, the model reinforces its initial error, incorrectly concluding that right atrial pressure increases.

**reduction** in venous return and right atrial pressure, thus decisively ruling out 'Option A'. Building on this correct premise, it re-evaluates the alternatives and correctly deduces that to maintain cardiac output, the body undergoes reflex tachycardia. This shortens the cardiac cycle, making a 'reduction in diastolic filling time' the most immediate and consistent physiological response.

- **Final Conclusion:** The model correctly arrives at answer 'E'.

---

**Right Case in Internalization of Reflection Capability Ablation Study(Q1)**

## Thinking
I'm analyzing the hemodynamic effects of a 47-year-old man with a mid-systolic murmur suggestive of aortic stenosis standing from a supine position. Standing causes blood to pool in the lower extremities due to gravity, reducing venous return and preload, particularly to the right heart. This could lower right atrial pressure. However, I initially consider that compensatory mechanisms, like sympathetic activation, might increase venous tone, potentially raising right atrial pressure (option A). Alternatively, reflex tachycardia from reduced venous return could shorten the cardiac cycle, reducing diastolic filling time (option E). Options B and C, involving increased left-sided pressures, seem unlikely, as reduced venous return lowers preload. Option D, about pacemaker potential, appears unrelated to immediate hemodynamic changes.

## Critical Reflection
My initial focus on increased right atrial pressure (option A) due to compensatory venous tone was flawed. The primary effect of standing is blood pooling in the lower extremities, directly reducing venous return and right atrial pressure, making option A incorrect. Options B and C are also incorrect, as decreased venous return lowers left ventricular end-diastolic and pulmonary capillary wedge pressures. Option D remains irrelevant, as it pertains to electrophysiological changes, not hemodynamics. Option E, a reduction in diastolic filling time, aligns with reflex tachycardia triggered by reduced venous return, which shortens diastole to maintain cardiac output. This is the most immediate and consistent physiological response to standing.

### Conclusion
Standing reduces venous return due to gravity, decreasing preload and triggering reflex tachycardia to maintain cardiac output. This shortens the cardiac cycle, reducing diastolic filling time. My initial consideration of increased right atrial pressure overlooked the primary effect of decreased venous return. The murmur's presence (suggestive of aortic stenosis) is secondary, as the question focuses on general hemodynamics. Thus, the correct answer is option E, reflecting the physiological response to standing.

**Final Answer: E**

---

Figure 9: Meaningful reflection chain for Q1 from the Med-REFL fine-tuned model. The model successfully identifies its flawed initial assumption and corrects its reasoning to arrive at the correct answer.

**Case Summary**   This case vividly demonstrates that Med-REFL training elevates the model from a state of 'invalid reflection'—where it reinforces errors—to one of 'meaningful reflection' that can identify and rectify its own logical fallacies. The model learns to grasp the principal clinical mechanism rather than getting lost in secondary effects, proving that Med-REFL successfully internalizes a robust self-correction capability.

## G.2 CONSTRUCTING REFLECTIVE PREFERENCE VIA DPO PAIRS

A core mechanism of Med-REFL is the construction of high-quality preference data, consisting of a 'Chosen' path of effective reflection and a 'Rejected' path of ineffective reflection. This case study presents a typical DPO pair used to train the model how to navigate a complex case with contradictory clinical information (Figure 10).

---

### Question(Q2)

A 51-year-old woman presents to your office with 2 weeks of fatigue and generalized weakness. She has a past medical history of diabetes, hypertension, and hyperlipidemia. She was recently diagnosed with rheumatoid arthritis and started on disease-modifying therapy. She states she has felt less able to do things she enjoys and feels guilty she can't play sports with her children. Review of systems is notable for the patient occasionally seeing a small amount of bright red blood on the toilet paper. Laboratory studies are ordered as seen below.
Hemoglobin: 12 g/dL
Hematocrit: 36%
Leukocyte count: 7,700/mm³ with normal differential
Platelet count: 207,000/mm³
MCV: 110 fL
Which of the following is the most likely etiology of this patient's fatigue?
Options:
A. Depression
B. Gastrointestinal bleed
C. Iron deficiency
D. Medication side effect
E. Vitamin B12 deficiency

---

Figure 10: The clinical scenario for Question 2, presenting contradictory clues of macrocytic anemia and signs of a gastrointestinal bleed.

---

### Question(Q3)

A 68-year-old male comes to the physician for evaluation of right flank pain. He has a history of diabetes and peripheral artery disease. His blood pressure is 160/90 mm Hg. Physical examination shows abdominal tenderness and right flank tenderness. An ultrasound shows dilation of the right ureter and renal pelvis. Which of the following is the most likely underlying cause of this patient's condition?
Options:
A. Renal artery stenosis
B. Benign prostatic hyperplasia
C. Diabetic nephropathy
D. Common iliac artery aneurysm
E. Urethral stricture

---

Figure 11: The clinical scenario for Question 3, a complex case of unilateral hydronephrosis in a patient with significant vascular comorbidities.

**The 'Rejected' Path (Ineffective Reflection)** This path demonstrates a confused reasoning process that fails to integrate all clinical clues (Figure 12).

- **Chain-of-Thought:** The model initially notes the high MCV but becomes entangled during its reflection. It fixates on the 'bright red blood,' linking it to a gastrointestinal (GI) bleed. However, it fails to resolve a core contradiction: a GI bleed typically causes iron deficiency anemia, characterized by a **low MCV (microcytic anemia)**, which is the opposite of the patient's lab finding. The reflection is ineffective, leading the model to misinterpret its own steps and prioritize a single, salient symptom over contradictory laboratory evidence.

- **Final Conclusion:** Unable to reconcile the conflict, the model incorrectly chooses 'B. Gastrointestinal bleed', misled by a single symptom while ignoring a key diagnostic result.

**The 'Chosen' Path (Effective Reflection)** This path demonstrates a superior reasoning process that logically integrates all available data (Figure 13).

## Reject Case in Med-REFL(Q2)

## Thinking

We have a scenario here about a 51-year-old woman who's been experiencing fatigue and generalized weakness for two weeks. Let me think... she has a history of diabetes, hypertension, and hyperlipidemia, and she was recently diagnosed with rheumatoid arthritis and started on some disease-modifying therapy. That's a lot of medical history to consider. She also mentioned feeling guilty because she can't play sports with her kids, which could hint at depression, but let's not jump to conclusions just yet.

Hold on, there's also this detail about seeing a small amount of bright red blood on the toilet paper. That's definitely important. Now, let's look at her lab results: hemoglobin is 12 g/dL, hematocrit is 36%, leukocyte count is 7,700/mm³, platelet count is 207,000/mm³, and MCV is 110 fL.

Oh right, the MCV being 110 fL is significant. A high MCV usually points to macrocytic anemia, not microcytic. So, this isn't typical of iron deficiency, which would cause microcytic anemia. This triggers a thought about vitamin B12 deficiency, which is a common cause of macrocytic anemia.

Let me think through the other options. Depression could certainly cause fatigue and a lack of interest in activities, but the lab results and the presence of bright red blood suggest something more specific. Gastrointestinal bleeding could explain the bright red blood, but it usually leads to iron deficiency anemia, which would show a low MCV, not a high one. Medication side effects are always a possibility, especially with new treatments for rheumatoid arthritis, but they don't typically cause macrocytic anemia.

OK, that makes sense. The combination of macrocytic anemia (high MCV) and the patient's symptoms, along with her medical history, strongly suggests vitamin B12 deficiency. This makes the most sense given the lab values and the clinical presentation.

Well, wrapping it all up, the most likely etiology of this patient's fatigue is vitamin B12 deficiency. Option E.
Hold on, I want to make sure I didn't screw up.
Alright, let's take a step back and reflect on my initial thoughts. I initially considered depression as a significant factor because of the patient's reported fatigue and reduced ability to engage in activities. However, I now realize that I might have been too quick to jump to this conclusion without thoroughly evaluating the other clinical findings.

The presence of bright red blood on the toilet paper was noted, but I didn't give it enough weight. This is a critical piece of information that strongly suggests a gastrointestinal bleed. I also misinterpreted the MCV value of 110 fL as indicating microcytic anemia, which is incorrect. A high MCV actually indicates macrocytic anemia, which is often associated with vitamin B12 deficiency or liver disease, not iron deficiency.

I overlooked the patient's medication history, particularly the use of metformin, which can lead to vitamin B12 deficiency. This is a significant oversight because it could directly contribute to her macrocytic anemia and fatigue.

Additionally, I didn't fully appreciate the significance of the gastrointestinal symptoms in the context of her anemia. Gastrointestinal bleeding can indeed cause anemia, but it typically leads to iron deficiency anemia, which would present with a low MCV, not a high one. The presence of bright red blood on the toilet paper, combined with the macrocytic anemia, should have led me to consider a different cause, such as a gastrointestinal bleed that might be causing acute blood loss.

In summary, my initial approach was flawed because I didn't integrate all the clinical findings and laboratory results effectively. I need to be more systematic in my evaluation, especially when dealing with complex cases with multiple potential causes.

## Conclusion

After reflecting on my initial thoughts, I realize that the most likely etiology of this patient's fatigue is a gastrointestinal bleed. The presence of bright red blood on the toilet paper, combined with the macrocytic anemia (MCV 110 fL), strongly suggests an acute gastrointestinal bleed. While vitamin B12 deficiency is a valid consideration, especially given her use of metformin, the immediate and significant finding of bright red blood points more directly to a gastrointestinal issue.

To summarize, the patient's symptoms and lab results, particularly the high MCV and the presence of bright red blood, make a gastrointestinal bleed the most likely cause of her fatigue. Therefore, the correct answer is B: Gastrointestinal bleed. This conclusion is based on a more thorough integration of the clinical findings and a better understanding of the implications of the lab results.

Figure 12: The 'Rejected' reasoning path for Q2. The model's reflection fails to reconcile the conflict between a GI bleed and macrocytic anemia, leading to an incorrect conclusion.

- **Chain-of-Thought:** The model identifies both the 'bright red blood' and the 'high MCV'. Crucially, during reflection, it clearly articulates the contradiction: a GI bleed would typically present with a low MCV, making it an unlikely cause for this patient's macrocytic anemia. This recognition is the key to correcting the reasoning path. The model then shifts its focus to a previously overlooked piece of information: the 'recent initiation of disease-modifying therapy' for rheumatoid arthritis. It correctly reasons that certain DMARDs are known to cause megaloblastic anemia, which manifests with an elevated MCV.

- **Final Conclusion:** This line of reasoning perfectly explains all the key clinical features: fatigue (from anemia) and a high MCV (from a medication side effect), while treating the hematochezia as a secondary or co-existing issue. The model thus arrives at the most comprehensive and correct answer: 'D. Medication side effect'.

**Case Summary**   This DPO pair explicitly teaches the model that when faced with conflicting clinical clues, superior reasoning involves finding a diagnosis that explains **all** key data points, rather than cherry-picking one and ignoring others. Through such contrastive learning, the model develops a preference for effective reflection that resolves clinical contradictions and integrates complex information holistically.

G.3   REFINING THE REASONING OF A SOTA MODEL (HUATUO-O1)

This case study demonstrates that Med-REFL can refine and enhance the reasoning process of even a specialized SOTA medical model like Huatuo-o1. It shows a shift from reasoning based on 'statistical prevalence' to a more precise diagnostic process based on 'patient-specific evidence' for the case presented in Figure 11.

**Huatuo-o1 Before Med-REFL Fine-Tuning**   The baseline model correctly identifies the relevant differential diagnoses but falters in its final decision-making, opting for the statistically common but less clinically fitting diagnosis (Figure 14).

- **Chain-of-Thought:** The model correctly interprets the right-sided hydronephrosis as a sign of urinary obstruction. It identifies two strong possibilities: BPH, a very common cause of obstruction in older men, and a common iliac artery aneurysm compressing the ureter, which is relevant given the patient's significant vascular history (PAD). This demonstrates a strong initial diagnostic capability.

- **Flawed Decision-Making:** Despite identifying the aneurysm as a compelling possibility, the model defaults to the more 'common' diagnosis in its final response. It reasons that an aneurysm is 'less commonly associated with direct ureteral compression compared to BPH'. This decision relies more on general prevalence than on the specific, strong vascular risk factors presented in this particular case.

- **Final Conclusion:** The model selects 'B. Benign prostatic hyperplasia'.

**Huatuo-o1 After Med-REFL Fine-Tuning**   After fine-tuning, Huatuo-o1 demonstrates sharper clinical acumen and the ability to correctly weigh competing evidence, as illustrated in its improved reasoning (Figure 15).

- **Chain-of-Thought:** The model's initial analysis is similar, identifying both BPH and an aneurysm as possibilities. The key difference lies in its **deeper critical evaluation** of the BPH diagnosis. It correctly critiques this option by noting that BPH typically causes bladder outlet symptoms rather than unilateral flank pain.

- **Optimized Decision-Making:** The fine-tuned model gives appropriate weight to the patient's **history of peripheral artery disease** as a powerful piece of evidence. It confidently connects this vascular history to the plausibility of an aneurysm, which would perfectly explain the external compression of the right ureter leading to unilateral hydronephrosis. This diagnosis integrates the patient's age, hypertension, PAD, and specific ultrasound findings into a single, coherent explanation.

- **Final Conclusion:** The model confidently selects 'D. Common iliac artery aneurysm', a more precise diagnosis rooted in the patient's specific clinical evidence.

## Chosen Case in Med-REFL(Q2)

## Thinking

We have a scenario here about a 51-year-old woman who's been experiencing fatigue and generalized weakness for two weeks. Let me think... she has a history of diabetes, hypertension, and hyperlipidemia, and she was recently diagnosed with rheumatoid arthritis and started on some disease-modifying therapy. That's a lot of medical history to consider. She also mentioned feeling guilty because she can't play sports with her kids, which could hint at depression, but let's not jump to conclusions just yet.

Hold on, there's also this detail about seeing a small amount of bright red blood on the toilet paper. That's definitely important. Now, let's look at her lab results: hemoglobin is 12 g/dL, hematocrit is 36%, leukocyte count is 7,700/mm³, platelet count is 207,000/mm³, and MCV is 110 fL.

Oh right, the MCV being 110 fL is significant. A high MCV usually points to macrocytic anemia, not microcytic. So, this isn't typical of iron deficiency, which would cause microcytic anemia. This triggers a thought about vitamin B12 deficiency, which is a common cause of macrocytic anemia.

Let me think through the other options. Depression could certainly cause fatigue and a lack of interest in activities, but the lab results and the presence of bright red blood suggest something more specific. Gastrointestinal bleeding could explain the bright red blood, but it usually leads to iron deficiency anemia, which would show a low MCV, not a high one. Medication side effects are always a possibility, especially with new treatments for rheumatoid arthritis, but they don't typically cause macrocytic anemia.

OK, that makes sense. The combination of macrocytic anemia (high MCV) and the patient's symptoms, along with her medical history, strongly suggests vitamin B12 deficiency. This makes the most sense given the lab values and the clinical presentation.

Well, wrapping it all up, the most likely etiology of this patient's fatigue is vitamin B12 deficiency. Option E.
Wait a minute, I'm not sure about that. Let me confirm.
Okay, let me take a step back and reflect on my initial thinking. I initially considered the bright red blood on the toilet paper as a significant clue pointing towards a gastrointestinal bleed, which I thought might be causing iron deficiency anemia. However, I overlooked a crucial detail: the MCV value of 110 fL, which indicates macrocytic anemia, not microcytic anemia. Iron deficiency anemia typically presents with a low MCV, so that was a major misstep in my reasoning.

Additionally, I didn't fully consider the patient's recent medication history. Starting disease-modifying therapy for rheumatoid arthritis can introduce new side effects, and some of these medications, like sulfasalazine, are known to cause megaloblastic anemia. This oversight in considering the medication side effects was a significant gap in my initial approach.

I also briefly considered depression, which can certainly cause fatigue and a lack of interest in activities, but the lab results and the specific nature of the anemia (macrocytic) pointed more strongly towards a physiological cause rather than a purely psychological one.

Reflecting on this, I realize that I should have integrated the MCV value more thoroughly into my differential diagnosis and considered the recent medication changes more carefully. This would have led me to the correct conclusion much sooner.

So, the key takeaway is that I need to pay closer attention to all available clinical data, especially lab results, and consider the full context of the patient's recent medical history and treatments. This holistic approach would help me avoid overlooking critical details and lead to a more accurate diagnosis.

## Conclusion

After reflecting on my initial thoughts and identifying the errors, I now have a clearer understanding of the patient's condition. The key points are the macrocytic anemia (MCV of 110 fL) and the recent initiation of disease-modifying therapy for rheumatoid arthritis. These factors strongly suggest that the most likely etiology of her fatigue is a medication side effect, specifically a medication-induced megaloblastic anemia.

The patient's recent use of sulfasalazine, a known cause of megaloblastic anemia, fits well with the lab results showing macrocytic anemia. While depression and gastrointestinal bleeding are valid considerations, they don't align as closely with the specific lab findings.

Therefore, the most likely etiology of this patient's fatigue is a medication side effect, and the correct answer is option D: Medication side effect. This conclusion integrates all the available clinical and laboratory data, providing a comprehensive and accurate diagnosis.

Figure 13: The 'Chosen' reasoning path for Q2, which forms the positive example in the DPO pair. The model correctly identifies the central contradiction and integrates the medication history to arrive at the correct diagnosis.

### Wrong Case in Med-REFL Huatuo-o1(Q3)

## Thinking

Alright, let's see what's going on with this patient. We've got a 68-year-old man who's dealing with some pretty uncomfortable right flank pain. He also has a history of diabetes and peripheral artery disease, which could complicate things a bit. His blood pressure is up there at 160 over 90, so we're definitely looking at hypertension.

Now, when I hear about flank pain and tenderness, my mind jumps to kidney issues. The ultrasound showing dilation of the right ureter and renal pelvis makes me think something's blocking the flow of urine out of that kidney. That's usually a red flag for hydronephrosis, where the kidney swells because it can't drain properly.

Okay, so why would that happen? In older men, one common culprit is benign prostatic hyperplasia, or BPH. It's like the prostate gets all enlarged and squishes the ureters, causing backup and swelling. But wait, this guy's got diabetes and PAD, so maybe we should consider other possibilities too.

Peripheral artery disease makes me think about big vessel problems, but does it relate here? Not directly, unless it somehow affects the arteries near the kidneys or bladder. Hmm, what else?

Renal artery stenosis could mess with kidney function, sure, but it doesn't really explain the dilation of the ureter. It's more about reduced blood flow than blockage. So, probably not that.

Diabetic nephropathy? Nah, that's more about damage to the kidney itself, not so much about causing a blockage in the urinary tract. And a urethral stricture would affect the lower part of the urinary system, not the ureter.

Oh, what about a common iliac artery aneurysm? Now that's interesting. If that thing balloons up, it might press on the ureter and cause obstruction. That could totally lead to hydronephrosis, especially if it's pressing from the side or back.

So, considering everything, while BPH is a usual suspect for these kinds of symptoms, the vascular history and the specific location of the dilation make me lean towards the idea of a common iliac artery aneurysm being the troublemaker here. It's less common, but it fits the picture better with his overall health profile.

Yeah, I'm feeling confident about this. The combination of his age, hypertension, and vascular issues points toward the aneurysm as the most likely reason for his current situation.

## Final Response

Based on the details provided, the most likely underlying cause of this patient's condition is **B: Benign prostatic hyperplasia**. While the patient's history of diabetes and peripheral artery disease adds complexity to the case, the key symptom of right flank pain accompanied by dilation of the right ureter and renal pelvis strongly suggests an obstruction in the urinary tract. Benign prostatic hyperplasia (BPH) is a common cause of such obstructions in older men, as it can compress the ureters and lead to hydronephrosis. Although a common iliac artery aneurysm is a possibility due to the vascular history, it is less commonly associated with direct ureteral compression compared to BPH. Therefore, the clinical presentation aligns more closely with BPH as the underlying cause.

Figure 14: The reasoning chain of the baseline Huatuo-o1 model for Q3. While identifying the correct differential, it defaults to the statistically common but less fitting diagnosis of BPH.

## Right Case in Med-REFL Fine-Tuned Huatuo-o1(Q3)

## Thinking

Alright, let's see what's going on with this patient. We've got a 68-year-old man who's dealing with some pretty uncomfortable right flank pain. He also has a history of diabetes and peripheral artery disease, which already sets off a few alarms about potential vascular issues. His blood pressure is up there at 160 over 90, so we're definitely looking at hypertension.

Now, when I hear flank pain and hypertension together, my mind jumps to something like kidney stones or maybe pyelonephritis. But wait, the ultrasound shows that his ureter and renal pelvis are dilated. That's interesting because it suggests there's some kind of blockage or backup happening in the urinary tract.

Okay, let's think about what could be causing this. With his age and symptoms, benign prostatic hyperplasia (BPH) is a common culprit for urinary obstruction in men. It can lead to bladder outlet obstruction, which might cause urine to back up into the kidneys, leading to hydronephrosis. But hold on, BPH usually causes bladder issues more than flank pain directly. Hmm, maybe not the best fit here.

Renal artery stenosis? Well, that's more about reducing blood flow to the kidney itself, often causing high blood pressure due to activation of the renin-angiotensin system. It doesn't really explain the dilation of the ureters and renal pelvis, though. So, probably not our guy.

Diabetic nephropathy? Nah, that's more about chronic damage to the kidney tissue itself, not really about physical obstructions or blockages. Plus, it wouldn't typically cause these specific ultrasound findings.

What about a common iliac artery aneurysm? Now that's intriguing. If one of those big arteries near the pelvis gets all swollen, it could press on the ureter and cause a backup, leading to hydronephrosis. This would match the dilation seen on the ultrasound. And since he's got peripheral artery disease, he's already got some vascular issues going on, making this possibility more plausible.

Urethral stricture? Hmm, that's more about narrowing further down the line, after the bladder, so it wouldn't cause dilation of the ureter and renal pelvis. It's less likely to be the main issue here.

So, putting it all together, while BPH is a usual suspect for urinary problems in older guys, the combination of his hypertension, peripheral artery disease, and the specific ultrasound findings makes me lean towards the idea of a common iliac artery aneurysm compressing the ureter. Yeah, that seems to tie everything together nicely.

## Final Response

Based on the details provided, the most likely underlying cause of this patient's condition is **D: Common iliac artery aneurysm**. The key factors pointing to this diagnosis include the combination of right flank pain, hypertension, peripheral artery disease, and the ultrasound finding of dilation of the right ureter and renal pelvis. These suggest a possible compression of the ureter by an aneurysm, rather than a typical obstructive urological cause such as benign prostatic hyperplasia. While BPH is common in elderly men and can cause urinary retention, it generally leads to bladder-related symptoms rather than direct ureteral dilation. In contrast, a common iliac artery aneurysm can exert external pressure on the ureter, resulting in hydronephrosis, aligning well with the clinical presentation observed in this case.

Figure 15: The refined reasoning chain of Huatuo-o1 after Med-REFL fine-tuning for Q3. The model demonstrates superior clinical acumen by critiquing the common diagnosis and correctly prioritizing patient-specific evidence.

**Case Summary**   Med-REFL refines Huatuo-o1's reasoning capability, elevating it from a model that knows 'what is common' to a clinical reasoner that understands 'what is most likely, given this specific patient's evidence.' This demonstrates that the framework enhances the model's ability to weigh evidence appropriately, leading to more accurate and logically sound diagnostic decisions.

# H   PROMPTS IN MED-REFL

## H.1   PROMPTS IN TREE OF THOUGHT PATH SEARCHING

Firstly, Llama3.1-8B start exploring the first step through prompt in Prompt H.1, then for the intermediate steps, Llama3.1-8B based on the previous steps and question generate the next step from prompt in Prompt H.1. After each step generation, we determine whether this step is a leaf node by utilizing prompt in Prompt H.1.

---

**Prompt for Generating Initial Tree Search Steps**

You are a medical expert specializing in USMLE exam questions.

---

**Your Task:**
Your task is to suggest three different **INITIAL** analytical steps for approaching a given medical question. Each approach should start from a unique analytical perspective and be specific and actionable.

**Important Rules:**

- This is an exam scenario; all patient information is **ONLY** in the question.

- Provide **ONLY** the FIRST step for each different approach.

---

**Input Format:**

```
QUESTION: [USMLE question]
```

**Output Format:**
Present your response in a single, valid JSON object with three keys: `step1`, `step2`, and `step3`.

```json
{
  "step1": "First_analytical_step_based_on_the_key_
      ↪ information_provided_in_the_question",
  "step2": "Second_analytical_step_based_on_the_key_
      ↪ information_provided_in_the_question",
  "step3": "Third_analytical_step_based_on_the_key_
      ↪ information_provided_in_the_question"
}
```

**Prompt for Exploring Intermediate Nodes in Tree Search**

You are a medical expert specializing in USMLE exam questions.

**Your Task:**
Based on the given medical question and previous problem-solving steps, suggest the **next three** analytical steps. Your analysis should focus on interpreting provided symptoms, developing differential diagnoses, evaluating treatment options, and applying medical knowledge to existing findings.

**Important Rules:**

- This is an exam scenario; all patient information is **ONLY** in the question.

- Do **not** suggest new tests, examinations, or patient interactions.

- Focus on analyzing and interpreting **existing information only**.

**Input Format:**

```
QUESTION: [USMLE question]
HISTORY: [Previous steps]
LASTSTEP: [Most recent step]
```

**Output Format:**
Present your response in a single, valid JSON object with three keys: `step1`, `step2`, and `step3`.

```
{
  "step1": "First_analytical_next_step",
  "step2": "Second_analytical_next_step",
  "step3": "Third_analytical_next_step"
}
```

---

**Prompt for Evaluating Sufficiency and Terminating a Thought Chain**

You are a medical expert specializing in United States Medical License exam questions.

---

**Your Task:**
You will be provided with a medical question, answer options, previous steps, and the current step. Your task is to determine if the given information is **sufficient** to answer the question.

---

**Input Format:**

```
1. Question: [A medical question from the US Medical License
   ↪ exam]
2. Options: [List of possible answer choices]
3. History: [Previous problem-solving steps conducted]
4. Step: [Current problem-solving step]
```

**Output Format:**
Your output **must** be a single, valid JSON object with three keys: `reason`, `decision`, and `answer`.

- **reason**: Explain your thought process.

- **decision**: Write "yes" if sufficient, "no" otherwise.

- **answer**: If "yes", provide the correct option. If "no", write "None".

```
{
  "reason": "Your_detailed_explanation",
  "decision": "yes" or "no",
  "answer": "Chosen_option_or_None"
}
```

## H.2 PROMPTS FOR BUILDING COMPLEX REASONING PROCESSES

After getting the first-person reflect and thinking from prompt in Prompt H.2, these two texts will be rearrange with $r$ from prompt in Prompt H.2 again to form the complex mid-reasoning reflection CoT. The post-reasoning reflection CoT could be directly formed by prompt in Prompt H.2

**Prompt for Generating a First-Person Chain-of-Thought**

Your task is to create a conversational, first-person chain of thought that sounds like a biomedical student thinking aloud while solving a problem.

**Instructions:**

1. **Start** with one of these natural conversation starters:
    - "Okay, so I've got this question about..."
    - "Well, let me see... this is asking about..."
    - "Hmm, interesting... we're looking at a problem about..."
    - "So here's what we've got... a question concerning..."
    - "Right, I need to figure out something about..."

2. **Continue** the thought process using casual, thinking-out-loud language (e.g., "let me think...", "wait a minute...", "oh right...").

3. **Maintain** a biomedical student's perspective with appropriate terminology.

4. **Show** natural pauses and realizations (e.g., "Ah, now I see...").

5. **Follow** the logical steps from the original reasoning path provided.

6. **Conclude** with a casual but confident summary (e.g., "So yeah, putting it all together...").

Write as if you're recording your actual thought process, including moments of reflection and connection-making.

**Prompt for Generating Educational Reflection**

You are a medical education assistant working with medical students and healthcare professionals. Your task is to analyze clinical reasoning processes and provide educational reflections on problem-solving approaches.

---

**Input Format:**
For each case, you will receive:

```
[Question] A clinical scenario with a multiple-choice
    ↪ question
[Public_Steps] The common initial reasoning steps
[Wrong_Steps] A step in the reasoning process that contains
    ↪ errors
[Right_Steps] The correct approach that should have been
    ↪ taken
```

**Task and Output Requirements:**
Generate a concise **Reflection**. Your reflection **must** include the following, without directly referencing the bracketed labels like "$[Wrong_{Steps}]$" or "$[Right_{Steps}]$":

1. A clear identification of the **flaws** in the initial thinking.

2. Analysis of what key information was **overlooked, misinterpreted, or not properly weighted**.

3. Identification of any **missing domain knowledge** or principles that led to the error.

4. A logical bridge explaining how recognizing these issues leads to the **correct approach**.

5. If applicable, mention any clinical **guidelines or best practices** that support the correct approach.

Keep your reflection focused on the critical analytical insights that explain the transition from incorrect to correct reasoning.

---

**Prompt for Generating a First-Person CoT with Reflection**

You will create a first-person chain of thought that demonstrates a natural problem-solving process, maintaining a conversational tone throughout.

---

**Input Format:**
You will be provided with the following:

```
[Question] The problem to solve
[First Think Solution] Initial reasoning that contains errors
[Reflection] Analysis of the errors in the initial solution
[Correction] The correct approach and solution
```

**Output Format:**
Create a response with these **exact** section markers: <thinking>, <reflection>, and <conclusion>. Your language should flow naturally as if capturing an authentic thought process.
**<thinking>**
Present your initial approach based on the [First Think Solution]. Use first-person, thinking-aloud language. This section should show your reasoning process, including the errors that will be corrected later.
**<reflection>**
Reflect on your initial thinking. Incorporate the content from [Reflection] to explain, in a first-person voice, why your initial approach was flawed, what you overlooked, and what new insight leads to a better solution.
**<conclusion>**
Synthesize your reflection and initial thinking to arrive at the correct solution based on [Correction]. Summarize how your thinking has evolved and confidently present the final answer.

---

During the answer verification process, we did not simply use rule matching. Instead, we first used prompt in Prompt H.2 to extract results from the model's responses through LLM, and then proceeded with answer comparison.

In the ablation study Section 4.1 , to verify our proposed hypothesis, we forced the model to conduct step-by-step reflection using the prompt in Prompt H.2. Then, using the prompt in Prompt H.2, we forced the model to reflect after thinking.

**Prompt for Extracting the Final Answer Option from Reasoning**

You are tasked with extracting and categorizing the chosen option from a medical reasoning text.

**Your Task:**
Given a text containing detailed reasoning and a final choice, your task is to analyze the text, identify the explicitly stated chosen answer, and extract only the corresponding letter.

**Input Format:**
A reasoning text that concludes with a chosen answer.

**Rules:**

- Analyze the text to find phrases indicating a final choice (e.g., "Therefore, the answer is", "The correct answer is").

- Extract **only** the single letter of the chosen option (A, B, C, D, or E).

- If the text does not clearly indicate a final choice, you **must** return 'None'.

- Ignore any additional text, numbers, or explanations surrounding the answer letter.

**Output Format:**
Your output **must** be in the exact format: `Answer:X`, where X is the letter of the chosen option or 'None'.

**Examples:**

```
Input: "...Therefore, the answer is D: ..."
Output: Answer:D

Input: "...The correct answer is 3 ... Options:'A':5,'B':3 "
Output: Answer:B

Input: "..., the most likely answer is \\boxed{C}."
Output: Answer:C

Input: "...While A is plausible,  B also has merits..."
Output: Answer:None
```

**Prompt for Standard Step-by-Step Evaluation**

Please think step by step to answer the following multiple-choice questions.

**Output Format:**
Ensure your response concludes with the correct option in the following exact format on a new line:
`The answer is X(Option Letter).`

---

**Prompt to Force Step-by-Step Thinking**

You are a medical expert solving USMLE exam questions.

---

**Instructions:**
When given a multiple-choice medical question:

1. Focus on the core medical concepts in the question.

2. Use your knowledge of pathophysiology, diagnosis, and treatments.

3. Analyze each answer option methodically.

4. Select the most appropriate answer.

**Output Format:**
You **must** respond in this exact format, using the specified section headers.
### Thinking
[Think step-by-step through the question as if you are taking an exam. Identify key information and analyze each option. If the question seems difficult, double-check your reasoning.]
### Conclusion
[Summarize the key points from your thinking and clearly state which answer option is correct.]

---

Keep your reasoning clear and concise. Always provide a definitive answer choice.

---

**Prompt to Force Reflection After Thinking**

You are a medical expert solving USMLE exam questions.

---

**Instructions:**
When given a multiple-choice medical question:

1. Focus on the core medical concepts in the question.

2. Use your knowledge of pathophysiology, diagnosis, and treatments.

3. Analyze each answer option methodically.

4. Select the most appropriate answer.

**Output Format:**
You **must** respond in this exact format, using the specified section headers.
### Thinking
[Think step-by-step through the question as if you are taking an exam. Identify key information and analyze each option. If the question seems difficult, double-check your reasoning.]
### Critical reflection
[Take time to reflect on your Thinking. Consider if there are alternative explanations or important factors you might have overlooked. Challenge your initial assumptions and verify if your reasoning aligns with standard medical practice.]
### Conclusion
[Summarize the key points from your previous reasoning process and clearly state which answer option is correct.]

---

Keep your reasoning clear and concise. Always provide a definitive answer choice.

