# OpenReview forum: "Med-REFL: Enhancing Complex Reasoning via Fine-grained Self-Correction"
_ICLR.cc/2026/Conference — Submitted to ICLR 2026_

### Official Review · Reviewer_3fRk · 2025-10-31

**Soundness:** 2
**Presentation:** 3
**Contribution:** 2
**Rating:** 4
**Confidence:** 2

**Summary:**

This paper introduces Med-REFL, a novel framework for enhancing complex medical reasoning in large language models through fine-grained self-correction. The key innovation is a deterministic, structure-based evaluation method using tree-of-thoughts (ToT) exploration to automatically generate preference data for direct preference optimization (DPO). The framework addresses the verification bottleneck in medical reasoning by quantifying the value of corrective actions without requiring human labels or expensive process reward models. Experiments show consistent improvements across diverse models and benchmarks, with particularly strong gains on the MedQA dataset (+3.67% average) and GPQA (+3.53% average).

**Strengths:**

1. **Well-motivated problem formulation**: The paper clearly articulates why existing approaches (outcome-based RL, PRMs) are insufficient for medical reasoning and positions Med-REFL as a solution to the verification bottleneck.

2. **Comprehensive experimental validation**: Testing across 7 diverse models (instruction-tuned, reason-heavy, knowledge-heavy) and 7 benchmarks with consistent improvements demonstrates robustness.

3. **Strong ablation studies**: The experiments comparing Med-REFL against MCTS rollouts and GPT-4.1-as-judge (Table 3) provide compelling evidence for the structural assessment approach. The prompting experiments (Table 5) elegantly demonstrate internalized reflection capability.

4. **Practical impact**: Achieving SOTA in the 7-8B class and competing with 10-20B models (Figure 1d, Table 7) has clear practical value for resource-constrained deployment.

**Weaknesses:**

**Computational cost not addressed**: ToT exploration is computationally expensive. The paper lacks analysis of:
   - Wall-clock time for data generation compared to alternatives
   - Trade-offs between exploration breadth/depth and final performance
   - Scalability analysis - what happens with 10x or 100x more data?

**Scoring model creates confusion**: Table 1 claims Med-REFL is "Reward Model Free" but Section 2.3 introduces a 3B scoring model trained on v_sol. While smaller than a full PRM, this contradicts the "free" claim and adds training complexity not fully acknowledged.

**Reflection vs. reasoning gains unclear**: Table 10 shows both components help, but the interaction effects and optimal ratios (beyond the 1:2 empirical finding) lack theoretical justification. Why is 1:2 optimal?

**Limited failure analysis**: The paper focuses on improvements but doesn't analyze:
   - Which question types or medical domains see minimal or negative impact?
   - Are there systematic biases introduced by the ToT exploration?
   - Cases where reflection leads to overthinking/correct→incorrect changes

**Questions:**

see weakness

---

> ### Author Response · Authors · 2025-11-24
> **Response to Reviewer 3fRk   [1/2]**
>
> We sincerely thank the reviewer for the thoughtful assessment and for recognizing the "practical impact" of our work。
>
> ## **Response to Weakness 1: Computational Cost & Scalability**
>
> We acknowledge that the computational cost of Tree-of-Thoughts (ToT) is a valid concern. However, we would like to clarify that Med-REFL utilizes ToT exclusively for **offline data generation**, ensuring zero additional cost during the model's inference phase. Furthermore, our newly added analysis demonstrates that this offline process is highly cost-effective compared to existing paradigms.
>
> - **Offline Generation vs. Online Inference:** The ToT exploration is a one-time, offline process used solely to construct the preference dataset. The final Med-REFL model is a standard auto-regressive model that does not require tree search during deployment.
>
> - **High Cost-Efficiency (Table 6, Line 433):** As detailed in **Table 6**, generating our entire 33k preference dataset (including ToT rollout, error location, and reflection generation) took only **~83 GPU hours** on a single A100, costing approximately **$166**.
>
>     - In contrast, distillation from models like GPT-4/5 typically costs **>$2,500** in tokens, and training a heavy Reward Model for PPO often exceeds **100 GPU hours**. Med-REFL is explicitly designed to be a resource-efficient solution.
>
> - **Trade-offs in Exploration (Figure 5, Line 1547):** Regarding the trade-off between exploration breadth/depth and performance, our analysis in **Appendix F.9** shows that valid reasoning trajectories follow a long-tail distribution with a mean length of **3.63 steps**. Our search budget (Depth=8, Branch=3) captures nearly 100% of valid paths. Increasing the budget further yields diminishing returns as the "reasoning boundary" is determined by the model's capability, not just search depth.
>
> - **Scalability:** While we agree that scaling the dataset size by 10x-100x is an exciting direction for future work, we have demonstrated **domain scalability**. As shown in **Table 5,Line 413**, we successfully adapted the framework to a logic puzzle domain ("Puzzle-REFL") using only 100 seed questions, achieving significant gains (+9.59% on hard tasks). This proves the framework's scalability across domains without requiring massive initial data.
>
>
> ## **Response to Weakness 2: Scoring Model & "Reward Model Free" Claim**
>
> We apologize if the term "Reward Model Free" caused confusion. We use this term to distinguish our method from the standard RLHF (PPO) paradigm, which necessitates training a complex Reward Model on expensive **human preference data**.
>
> - **Clarification of Definition:** By "Reward Model Free," we mean Med-REFL does not require human preference data or the training of a heavy reward model for online policy optimization (e.g., PPO).
>
> - **Role of the Scoring Model:** The 3B scoring model in Section 2.3 acts strictly as a **lightweight, offline data filter**. It is trained on rule-based outcome labels ($v_{sol}$), not human preferences, and is used solely to rank trajectories during dataset construction8. It is **not** involved in the online training loop or inference, adding no complexity to the alignment process itself.
>
> - **Necessity and Effectiveness:** As analyzed in **Appendix F.3 (Figure 3, Line 1234)**, this lightweight scoring model is crucial for filtering high-quality reasoning paths. It significantly outperforms a simple multi-vote baseline (59.86% vs. 57.03%) 9, validating that our deterministic structural assessment ($v_{sol}$) provides a superior signal for data selection compared to statistical voting.

---

> ### Author Response · Authors · 2025-11-24
> **Response to Reviewer 3fRk      [2/2]**
>
> ## **Response to Weakness 3: Reflection vs. Reasoning Gains & Justification for 1:2 Ratio**
>
> We thank the reviewer for this insightful question. The 1:2 ratio is not arbitrary; it emerged from our empirical investigation into balancing **correction capability** with **inference stability**.
>
> - **The "Over-Reflection" Problem:** Initially, we trained models exclusively on Reflection Learning data. We observed that this caused the model to overfit to a "constant reflection" paradigm: the model would initiate a reflection step even for simple questions or when its initial reasoning was already correct. This not only wasted tokens but, in some cases, led to "False Correction," where the model doubted a correct initial thought due to learned uncertainty.
>
> - **Reasoning Data as a Regularizer:** To mitigate this, we introduced Reasoning Enhancement data (standard high-quality CoT without reflection) to act as a **regularizer**.
>
>     - **Function:** It teaches the model that "confident, direct reasoning" is often the optimal path for clear-cut problems. It stabilizes the model's distribution, preventing it from questioning every output.
>
>     - **Why 1:2 Ratio?** The capability to "detect errors and self-correct" is a more complex, harder-to-learn behavior than standard reasoning, thus requiring a stronger signal (Weight=2). The Reasoning data (Weight=1) provides just enough regularization to maintain efficiency and confidence without diluting the reflection signal. This balance is empirically supported by Appendix F2, Table 11 (Line 1217) , where deviating from this ratio degrades performance.
>
>
> ## **Response to Weakness 4: Limited Failure Analysis**
>
> We appreciate the suggestion to analyze failure modes, as understanding limitations is key to robust deployment. We have added a new analysis in **Table 4, Line394** to address this.
>
> - Quantitative Mechanism (Table 4, Line 394):
>
>     - One concern might be that the model simply changes answers randomly. Our analysis shows that Med-REFL increases the **Reflection Divergence Rate** (changing the answer after thinking) from 1.38% to 2.23%.
>
>     - Crucially, this is not noise: the **Correction Success Rate** (probability that a change leads to the correct answer) nearly doubles from **12.76% to 19.83%**. This confirms the model is learning directed error correction, not just stochastic instability.
>
> - **Qualitative Failure Modes:** We have categorized the primary failure cases into two types:
>
>     1. **False Correction (Over-Reflection):** In highly ambiguous cases, the model may over-scrutinize a correct initial intuition and "correct" it to a wrong answer. As mentioned in our response to W3, this highlights the trade-off between self-correction and confidence, which we mitigate by including Reasoning Enhancement data.
>
>     2. **Knowledge Boundaries (Hallucination):** We observed that Med-REFL enhances **logical consistency** but cannot generate **missing knowledge**. If the model fundamentally lacks a specific medical fact (e.g., a rare drug interaction), the reflection process cannot retrieve it. Instead, the model may hallucinate a plausible-sounding justification to support a wrong guess.

---

### Official Review · Reviewer_oh3a · 2025-10-31

**Soundness:** 3
**Presentation:** 3
**Contribution:** 2
**Rating:** 6
**Confidence:** 3

**Summary:**

This paper presents Med-REFL, a framework that teaches large reasoning models to self-correct without human or reward-model supervision. It evaluates reasoning trees deterministically—assigning step and action values—to build preference pairs between effective and invalid reflections, then fine-tunes via Direct Preference Optimization (DPO).

Across seven medical benchmarks and multiple 7–8B models (Llama-3.1, Huatuo-o1, AlphaMed, etc.), Med-REFL achieves consistent gains (+3–6 %), reaching SOTA among 8B models.
Ablations show deterministic scoring outperforms MCTS or GPT-4.1 judges, and reflection ability can be explicitly activated by prompting, suggesting genuine self-correction.

**Strengths:**

The paper tackles one of the central challenges in medical reasoning — the lack of reliable supervision for intermediate steps. This is a genuine bottleneck for developing trustworthy reasoning models in high-stakes domains, and the authors motivate it well.

The proposed method delivers noticeable and repeatable gains across a wide range of models and benchmarks, not just in one narrow setup. The results look robust, suggesting the approach generalizes beyond a single architecture or dataset.

Includes thoughtful and convincing ablations. The ablation studies are well designed and help clarify the role of each component in the framework. They make it much easier to trust that the improvements come from the core ideas rather than from implementation details or data artifacts.

**Weaknesses:**

Still somewhat tied to QA-style supervision. Even though the deterministic scoring idea is neat, it ultimately depends on having the correct final answer to judge whether a reasoning path is good or bad. That works well for multiple-choice or QA-type benchmarks, but it might not translate easily to more open-ended tasks like diagnosis summaries or free-form clinical explanations where there isn’t a single “right” answer. It would be helpful if the authors could comment on how their approach might generalize to those cases.

Compute cost isn’t very clear. The paper describes generating full Trees-of-Thought and evaluating every step, which sounds potentially expensive, but it doesn’t quantify how costly that actually is. A brief analysis of runtime, GPU hours, or scaling behavior would make it easier to judge how practical this is compared to standard fine-tuning or PRM methods.

The authors claim the framework is general, but all experiments are in the medical domain. Even one small example from another reasoning area—say scientific QA or legal reasoning—would help show that the method really transfers beyond the datasets it was built for.

The conceptual novelty is somewhat incremental. The proposed framework builds on ideas already explored in process reward models (PRMs) and StepWiser-like step-evaluation methods, where each reasoning step is scored or judged to guide model improvement.

**Questions:**

1. How sensitive is the framework to the quality of the base model’s reasoning? Could a weaker base model still generate useful reflection data?
2. Is the step-value metric robust to partial correctness or multi-label questions?
3. How much compute was required to generate the 33k DPO pairs?

---

> ### Author Response · Authors · 2025-11-24
> **Response to QA Dependence (W1) and Metric Robustness (Q2)   [1/3]**
>
> We thank the reviewer for the thoughtful assessment. In this first post, we address the concerns regarding our reliance on QA-style supervision and the mathematical robustness of our metric.
> We have updated a new version of the paper PDF; the descriptions of all figures and tables below can be found in the corresponding places in this new version.
>
>
> ### 1. Response to Weakness 1: QA-style Supervision & Generalization
>
> We acknowledge that Med-REFL currently relies on ground truth labels from QA tasks. However, we argue that this is a **methodological necessity for safety** rather than a limitation of capability, and our new experiments prove the learned skill is transferable.
>
> - **The Necessity of Verification:** In high-stakes domains like medicine, hallucination is unacceptable. QA tasks serve as a rigorous reasoning bottom line where logic must adhere to strict verifiable constraints. We believe models must master reflection in environments with clear Ground Truth before moving to open-ended tasks (e.g., diagnosis summaries) where verification is subjective. This insight is also shared by contemporary works such as Huatuo-o1, MedReason, and AlphaMed.
>
> - **Internalized Skill, Not Memorization:** The core value of Med-REFL is that it teaches the model **how to correct errors**, a cognitive skill that transcends the QA format.
>
>     - **Evidence of Genuine Correction:** In Table 4 (Line 393) , we quantified this mechanism. Med-REFL fine-tuning nearly **doubles the Correction Success Rate** (from 12.76% to 19.83%), proving the model is actively identifying and fixing logical fallacies, not just mimicking answer patterns.
>
>     - **Evidence of Transferability:** As detailed in Table 5 (Line 413), we applied the Med-REFL framework to a completely different domain: **Logic Puzzles (Knights and Knaves)**. The model trained _solely_ on medical QA data achieved a performance gain on logic puzzles (+2.98%), and applying our framework to puzzles yielded a +9.59% gain3333. This confirms the "reflection capability" generalizes beyond medical QA.
>
>
> ### 2. Response to Question 2: Robustness to Partial Correctness (Mathematical Proof)
>
> You asked if our step-value metric is robust to partial correctness or multi-label questions. **The answer is yes.**
>
> We can mathematically demonstrate that our Action Value formulation is invariant to the scoring granularity of leaf nodes.
>
> **Proof of Equivalence:**
>
> In our current implementation (Binary QA), the value of a leaf node $l$ is $r(l) \in \{+1, -1\}$. The step value $v_{step}(s)$ for an intermediate node $s$ is defined as the average outcome of its subtree leaves4:
>
> $$v_{step}(s) = \frac{1}{N_s} \sum_{l \in Leaves(s)} r(l) = \frac{N_{correct} - N_{incorrect}}{N_{total}} = \mathbb{E}[r|s]$$
>
> This is essentially a Monte Carlo estimate of the expected reward from state $s$.
>
> For partial correctness or multi-label tasks, we simply redefine the leaf reward $r'(l)$ to be a continuous variable $r'(l) \in [-1, 1]$ (e.g., normalized F1 score or accuracy ratio). The value of the intermediate node $s$ becomes:
>
> $$v_{\text{step}}^{\prime}(s) = \frac{1}{N_s} \sum_{l \in \text{Leaves}(s)} r^{\prime}(l) = \mathbb{E}[r^{\prime}|s]$$
>
> The Action Value $v_{act}(a_{s \to s'})$ (Eq. 4 in the paper 5) measures the expected improvement in value:
>
> $$v_{step}(s) - v_{step}(s') = \mathbb{E}[r|s] - \mathbb{E}[r|s']$$ ,
>
> and it is similar to $v_{sol}$ and $v_{rem}$
>
> **Conclusion:**
>
> Due to the linearity of the expectation operator, the structure of the Action Value calculation remains mathematically equivalent regardless of whether $r$ is binary $\{+1, -1\}$ or continuous $[-1, 1]$.
>
> Therefore, applying Med-REFL to multi-label tasks only requires replacing the binary leaf reward with a scalar metric (e.g., Jaccard index or F1 score), without any changes to the core algorithm.

---

> ### Author Response · Authors · 2025-11-24
> **Response to Computational Cost (W2, Q3) and Generalizability (W3)     [2/3]**
>
> In this second post, we provide the specific quantitative data requested regarding compute resources and address the concern about domain specificity with new experiments.
>
> ### 1. Response to Weakness 2 & Question 3: Computational Cost
>
> You rightly pointed out that generating ToT trajectories sounds potentially expensive and asked for quantification. **We have performed a detailed cost analysis in Table 6 Line 434, demonstrating that Med-REFL is highly cost-effective compared to alternatives.**
>
> - Specific Compute Data (Answering Q3):
>
>     As detailed in Table 6, our entire data generation pipeline for the 33k DPO pairs—including ToT rollout ($\approx 70h$), error location ($\approx 3h$), and reflection generation ($\approx 10h$)—required approximately 83 GPU hours on a single A100.
>
> - Economic Feasibility:
>
>     Based on standard cloud rental rates ($\approx$ \$2/hr), the total cost for constructing the dataset is roughly 166 dollar. The subsequent fine-tuning requires only 12 GPU hours.
>
> - Comparative Efficiency:
>
>     We highlight that this is a one-time, offline cost. In contrast, training effective Process Reward Models (PRMs) often necessitates training separate, parameter-heavy reward models (&gt;100 GPU hours), and distillation from proprietary models like GPT-5 involves prohibitive API costs (estimated &gt;\$2,500). Thus, Med-REFL offers a transparent and resource-efficient solution for the 7-8B model class.
>
> ### 2. Response to Weakness 3: Generalizability Beyond Medicine
>
> We appreciate your suggestion to test the framework on other reasoning tasks. To address this, we extended our evaluation to the **Knights and Knaves (K&K)** benchmark, a challenging **open-ended logic puzzle task** unrelated to medicine.
>
> As shown in **Table 5 Line 413**, we found compelling evidence of generalizability:
>
> - **Skill Transfer:** The model trained *solely* on our medical data (+Med-REFL) outperformed the base Llama3.1-8B on these logic puzzles (28.00% vs. 25.02%). This suggests the "self-correction capability" instilled by Med-REFL is a transferable cognitive skill, not just domain memorization.
>
> - **Methodological Transfer:** We applied the Med-REFL data generation pipeline to the puzzle domain ("Puzzle-REFL") using only 100 seed questions$^{11}$. This yielded a substantial performance jump to **34.61%** (+9.59% over base).
>
>
> These results confirm that while our paper focuses on medicine as a testbed for complex reasoning, the Med-REFL framework is domain-agnostic and can be effectively transferred to other logic-intensive domains.

---

> ### Author Response · Authors · 2025-11-24
> **Response to Conceptual Novelty (W4) and Model Sensitivity (Q1)        [3/3]**
>
> In this final post, we address the concern regarding conceptual novelty and the question about base model sensitivity.
>
> ### 1. Response to Weakness 4: Conceptual Novelty (Beyond PRMs)
>
> You noted that our framework builds on existing ideas like PRMs. While we acknowledge the lineage, we emphasize that **Med-REFL represents a fundamental shift from "Verification" to "Correction," and from "Discriminator" to "Generator."**
>
> - **Mechanism Difference (Action vs. State):** Traditional PRMs typically assign a scalar score to a state (step) to determine "is this step correct?" (Verification). Med-REFL, specifically through our **Action Value ($v_{act}$) metric** (Eq. 4), evaluates the _transition_ between states. We do not just learn "which step is bad"; we utilize the global tree structure to quantify **how much a corrective action improves the trajectory**.
>
> - **Outcome Difference (Generative Correction):** PRMs are mostly used as discriminators during inference (e.g., Best-of-N). In contrast, Med-REFL uses these structural valuations to construct DPO pairs that **fine-tune the generator itself**. This instills an internalized "self-correction" policy that persists even without the tree search at inference time (as proven by our prompting experiments in Section 4.1).
>
> - **Empirical Superiority:** As shown in **Table 3** of the main paper, Med-REFL outperforms strong baselines like "GPT-4-as-judge" and "MCTS-Rollouts." This empirically proves that our deterministic, structure-based assessment provides a superior learning signal compared to standard heuristic or statistical methods used in prior work.
>
>
> ### 2. Response to Question 1: Sensitivity to Base Model Quality
>
> You asked if a weaker base model could still generate useful data. We have investigated this extensively and found that **the base model quality defines the "Reasoning Boundary" for effective data generation.**
>
> - **The "Sweet Spot" Finding:** As detailed in our ablation study (**Appendix F.6 & Table 14, Line1316 ** of the main paper), we tested data generation using 3B, 8B, and 70B models:
>
>     - **3B Model (Too Weak):** Performance gain was minimal (+0.45%) because the model struggled to find _any_ correct reasoning path for difficult problems, leading to a lack of positive samples.
>
>     - **70B Model (Too Strong):** The gain was also lower (+0.86%) because the model rarely made the diverse, plausible errors needed for reflection. It effectively "knew too much" to need self-correction on this dataset.
>
>     - **8B Model (Optimal):** The 8B model provided the optimal balance (+1.73%), generating a rich set of both errors and successful corrections.
>
> - **Implication:** This confirms that Med-REFL is most effective when the problem difficulty lies within the model's **"Reasoning Boundary"** (discussed in **Figure 5 & 6 Line 1548**) — difficult enough to induce errors, but solvable enough to allow for self-correction via search.

---

### Official Review · Reviewer_83j9 · 2025-11-01

**Soundness:** 2
**Presentation:** 3
**Contribution:** 2
**Rating:** 4
**Confidence:** 3

**Summary:**

This paper introduces Med-REFL, a deterministic tree-of-thoughts framework that constructs step-wise preference data to DPO-train models, enabling fine-grained self-correction reasoning in medical fields.

**Strengths:**

1. Deterministic structural evaluation: This work utilizes step / solution / remaining value metrics and another action value $$v_{act}$$ to provide a clear and label-free signal to assess both states and corrective actions across the entire ToT search space.
2. Targeted reflection supervision via DPO: This work carefully designs a scheme to construct preference pairs that explicitly contrast effective versus invalid reflections, incentivizing self-correction ability of the  model rather than mere outcome accuracy.
3. Balanced training data: This work combines reflection-focused pairs with general CoT quality pairs, scored by a lightweight SFT scorer, to maintain broad reasoning proficiency while emphasizing error recovery.

**Weaknesses:**

1. Heuristic sensitivity: Performance may hinge on the quality of LLM-based tools ($E_{\pi}$, $C_{\pi}$, $M_{\pi}$) and hyperparameters (e.g., $\lambda_1$,$\lambda_2$,$\lambda_3$), which could vary across domains or models.
2. Limited algorithmic contribution: Med-REFL primarily leverages ToT to collect reasoning trajectories and uses carefully designed metrics to construct DPO preference pairs, thus porting self-reflection to the medical domain.

**Questions:**

Please refer to Weakness 1 & 2.
Here are some other questions:
1. In Table 2, the improvements of Pure-RL model are noticeably lower than that of other models, and there is even a performance drop on PubMedQA. Since the compared models are roughly similar in parameter size, could the authors explain what is the cause of such phenomenon? Why does your method appear to yield very limited gains on the Pure-RL model, while achieving the best improvements on the reason-heavy Huatuo-o1-8B?
2. In line 257, the authors mention introducing general-purpose data during training. What is the ratio between this general-purpose data and the reflection data, and is there a trade-off between performance on medical datasets and general-purpose benchmarks? If so, could the authors explain how they balanced the data mixture? Additionally, if the general-purpose data is removed, can the model—after training on well-organized DPO data—maintain its performance on general-purpose benchmarks?

---

> ### Author Response · Authors · 2025-11-21
> **Response to Weaknesses (Robustness & Novelty)      [1/3]**
>
> # Response to Heuristic Sensitivity and Robustness (W1)
> 1. **Deterministic Evaluation over Heuristics:**Our approach relies on a deterministic structural assessment of the entire ToT trajectory space to define the quality signal (e.g., $v_{step}$). This function is rule-based (counting correct/incorrect sub-paths) and independent of any LLM-as-a-Judge prompting, ensuring the underlying learning signal is stable.
> 2. **Robustness Analysis is Provided**: We conducted a thorough Sensitivity and Robustness Analysis in Figure5&6 (Line 1505) to empirically validate the stability of our method against internal fluctuations:
> - **Sampling Parameters**: We found that due to the nature of ToT (which aggregates multiple rollouts akin to a high Pass@K strategy), the framework remains robust against variations in random seeds and temperature.
> - **Search Budget**: While low search budget will decrease framework's performance, we confirm that our chosen search budget (Branching factor $B=3$, Depth $D=8$) is sufficient to capture nearly all natural reasoning lengths (near 100% cumulative coverage), avoiding premature truncation of the problem's Capability Boundary.
> - **Action Value Weights ($\lambda$)**: Crucially, we empirically validated the weights of the key metric, $v_{act}$ (Equation 4) , proving that our chosen ratio of $\lambda_1:\lambda_2:\lambda_3=2:1:2$ achieves the maximum separability between "Effective Actions" and "Invalid Actions" (minimum intersection area of 0.168) thus providing the most reliable training signal.
>
>
> # Response to Limited algorithmic contribution (W2)
>
> We appreciate the reviewer's comment. Our contribution extends beyond a simple application of these methods to the medical domain. Our work focuses on solving the critical verification bottleneck inherent to complex medical reasoning, a challenge fundamentally different from domains like mathematics where intermediate steps are easily verifiable.
>
> 1. Core Algorithmic Innovation: "Deterministic Reflection Signal Construction"
> The primary novelty lies in our deterministic, structure-based evaluation to automatically synthesize a high-quality preference signal for fine-grained reflection, eliminating the need for expensive human labels or a large, separate PRM. We move beyond simple outcome-based rewards and heuristic sampling by introducing the novel Action Value ($\mathbf{v_{act}}$) metric (Equation 4). This metric is specifically designed to quantify the value of a corrective action, enabling the direct comparison not just between a good path and a bad path, but fundamentally between effective reflection and invalid reflection
>
> 2. System Goal: Internalized, Fine-Grained Self-Correction
> Previous work on self-correction often relies on post-hoc reflection (model re-solves the problem after failure) or requires training computationally intensive Process Reward Models (PRMs) via complex multi-stage pipelines. * Med-REFL, by contrast, integrates this fine-grained process supervision signal into DPO training to internalize the self-correction capability into the model's parameters. * As confirmed by our ablations (Table 3 and Table 4), this results in a genuine, prompt-activatable reflection skill that allows the model to identify logical fallacies and execute valid self-corrections efficiently at inference time. * Our framework provides a unified, teacher model-free, reward model-free solution, a crucial step for scalable and cost-effective AI in high-stakes domains (Table 6, Line 433).

---

> ### Author Response · Authors · 2025-11-21
> **Response to Q1     [2/3]**
>
> # The reason for AlphaMed not meeting expectations
> - Possibility 1: The AlphaMed model has certain overfitting issues. It always attempts to analyze each option individually and then uses the process of elimination by comparing them to the question to solve problems, which is a form of overfitting to MCQ problems. Without deliberate prompt guidance, it lacks reflective capabilities on its own.
>
> - Possibility 2: Another possibility is that the scope of our model's core training questions is limited to problems where the PassK accuracy satisfies $0 < \text{Acc(Pass@K)} < 1$ (i.e., the reasoning boundary). This partially overlaps with the core training question scope of the RLVR model, which may lead to the Pure-RL model already having gained sufficient improvements during the RL stage, resulting in our method not achieving the expected gains.
>
> Conclusion: We will explore the specific situation in more detail in our future work. Regarding the performance of the AlphaMed model, we believe that possibility 1 accounts for a larger proportion.
>
> # Reasons for the decline in AlphaMed's PubMed QA
>
> PubMedQA is a dataset that heavily relies on factual knowledge. Since our method primarily optimizes the reasoning process, the overall improvement is not significant. Additionally, AlphaMed exhibits a certain degree of overfitting to the original patterns, which leads to the observed performance decline.

---

> ### Author Response · Authors · 2025-11-21
> **Response to Q2     [3/3]**
>
> # Q: Clarification on "General-Purpose Data," Data Ratio, and Trade-offs (Line 257).
>
> We appreciate the reviewer for pointing this out. We realize that the term “general-purpose data” in Line 257 may have caused confusion regarding our data sources. We would like to provide the following clarifications and insights:
>
> ### Clarification on Data Source (No External Domain Data Used)
> First, we wish to clarify that the "general-purpose reasoning data" (referred to as "Reasoning Enhancement data" in Sec 2.3 and Appendix E.2) does not come from non-medical general domains (e.g., GSM8K).
>
> ### Ratio and Ablation Results (Response to "What if removed?")
> The reviewer correctly identifies the importance of this data component.
>
> - **Optimal Ratio:** As detailed in Appendix F.2 (Table 11, Line1217), we conducted extensive experiments on the mixing ratio. We found that a 1:2 ratio (1 part Reasoning Enhancement : 2 parts Reflection Learning) yields the best performance.
> - **Impact of Removal:** As analyzed in Appendix F.3 (Table 13,Line1303), removing this "general-purpose" (Reasoning Enhancement) data leads to a significant performance drop.
> For Llama3.1-8B: accuracy drops from 65.74% (Full) to 60.85% (only Reflection data).
> This confirms that the model cannot maintain its peak performance if this data is removed.
>
> ### Why Do We Need "Reasoning Enhancement" Data?
> Why does adding standard reasoning paths boost the performance of a reflection-based model? During our pilot experiments we observed that training solely on reflection data introduced a bias we term "Universal Doubt" or "Over-Correction."
> - **The Issue: ** Without "Reasoning Enhancement" Data, the model learns a prior that preceding thoughts are always wrong (P(needs_correction) much higher ). This causes the model to reflexively "evaluate and correct" even simple, correct reasoning steps, often leading to hallucinated errors and wrong answers.
> - **The Solution:** The Reasoning Enhancement data serves as a "Correctness Anchor." It teaches the model that initial reasoning paths can be valid. This calibrates the model's confidence, enabling it to distinguish when to maintain its current path (Maintenance) and when to trigger reflection (Correction).
> Conclusion: By mixing these two data types, Med-REFL acts not just as a "corrector," but as a robust discriminator that can navigate the reasoning space more effectively, identifying high-value paths whether they come from the first attempt or a corrected one.

---

### Official Review · Reviewer_dN7n · 2025-11-02

**Soundness:** 3
**Presentation:** 3
**Contribution:** 2
**Rating:** 4
**Confidence:** 4

**Summary:**

Med-REFL is a teacher/PRM-free framework to instill fine-grained, activatable self-correction in medical LLMs. It explores tree-of-thought (ToT) trajectories per question and assigns deterministic, structure-based values to steps and actions using only final-answer supervision.It then constructs DPO preference pairs by contrasting effective with invalid reflections for reflection learning and high-quality with plausible-but-flawed reasoning for reasoning enhancement, using a lightweight scoring model. Fine-tuning on 33k pairs yields consistent gains across 7–8B models on MedQA, GPQA-M, MMLU-Pro, MedXpert, with strong ablations and evidence that reflection can be prompted post-training.

**Strengths:**

- Clear, deterministic valuation of steps enables process-level learning without PRMs or human labels.
- Explicitly trains “effective vs. invalid reflection,” shifting from best-path selection to error correction.
- Consistent improvements across diverse models/benchmarks; strongest on reasoning-heavy tasks.
- Reflection-activation study supports genuine internalization of self-correction.

**Weaknesses:**

- Valuation may be sensitive to ToT exploration distribution and generator choice, limited sensitivity analysis.
- Evaluation focuses on MCQs,vmodest gains on retrieval-heavy tasks; no process metrics (faithfulness, coherence) or open-ended tests.
- Fixed λ weights and 1:2 data mix lack broader sensitivity studies.
- preference data sourced mainly from MedQA.

**Questions:**

- Sensitivity of v_step/v_act to ToT parameters (branching/depth, seeds, temperatures) and λ weights?
- Any leakage controls between MedQA training-derived pairs and test split?
- Can you add open-ended/process evaluations (e.g., faithfulness, hallucination rates, calibration) and multistage cases?
- Is reflection text necessary, or does path preference alone suffice? What are compute/time costs for ToT/data construction?

---

> ### Author Response · Authors · 2025-11-21
> **Response to Sensitivity (Q1) and Data Leakage (Q2)         [1/3]**
>
> We thank the reviewer for the constructive feedback and for recognizing our framework's ability to "instill fine-grained, activatable self-correction" without human labels.
>
> # Answer for Sensitivity Concern (Q1)
> To address your concern about robustness, we have added a comprehensive sensitivity analysis (Appendix F.9, Line 1505). We explained that these hyperparameters affect the frame's performance by influencing the reasoning boundary[1]. "Reasoning boundary"—the zone where problems are difficult enough to induce errors (necessary for learning correction) yet solvable enough to find at least one correct path.
> ### **Stability across Temperature & Seeds (Line1517)**
> The ToT (controlled by seeds and temperature) rolling out strategy similarly to "Pass@K". Our analysis confirms that as long as the sampling parameters allow sufficient diversity ($T>0$), the probability of discovering a correct path for a problem within the model's capability boundary remains stable. Variations primarily shift which valid path is found first, rather than whether one is found4.
> ### **Impact of Search Budget (Branching/Depth) (Figure5, Line1529)**
> We acknowledge that low search budget would constrain the reasoning space. To validate our choice of $B=3, D=8$, we analyzed the step distribution of valid solutions in Figure 5, which shows a long-tailed distribution with a mean step count of 3.63 and a median of 36. So, our maximum depth of $D=8$ covers nearly 100% of the model's natural reasoning trajectories. Increasing depth yields diminishing returns, while decreasing it would artificially contract the capability boundary.
> ### **Sensitivity of $\lambda$ Weights (Figure6, Line1605)**
> We empirically validated our $\lambda$ ratio (2:1:2) by plotting the distribution of $v_{act}$ scores for "Right Actions" vs. "Wrong Actions" under different $\lambda$ ratios. We used the **Intersection Area (Overlap)** between the two distributions as a metric for separability (lower is better). As a result, the 2:1:2 ratio achieved the minimum intersection area (0.168) This confirms our setting provides the strongest signal to distinguish effective self-correction. Furthermore, this graph also illustrates that v_act is not sensitive to $\lambda$.
>
> # Answer for Data Leakage (Q2)
>
> ### **Strict Adherence to Official Splits**
> As detailed in **Line 323** and **Line 1056**, our preference dataset construction was performed **exclusively on the official MedQA-USMLE training set**. The test set was completely isolated and reserved solely for the final performance evaluation.
>
> ### **Evidence from OoD Generalization**
> If the performance gains were due to memorization or leakage, we would expect high scores on MedQA but poor performance on unseen benchmarks. However, **Table 2**(Line 270) shows that Med-REFL achieves robust gains across diverse **OoD** datasets , including **GPQA (+3.53%)** , **MedXpert (+4.71% on Reasoning)**  and **MMLU-Pro (+2.20%)**. These benchmarks have no overlap with MedQA, confirming that Med-REFL instills a transferable _reasoning and reflection capability_ rather than overfitting to specific questions.
>
> # Reference
> [1] Chen, Qiguang, et al. "Unlocking the capabilities of thought: A reasoning boundary framework to quantify and optimize chain-of-thought." Advances in Neural Information Processing Systems 37 (2024): 54872-54904.

---

> ### Author Response · Authors · 2025-11-21
> **Response to Open-ended Test, process evaluation (Q3&W2)     [2/3]**
>
> Thank you for your interest in the details of our framework.
>
> # An Open-ended Logic Puzzle Test (Table 5, Line 413)
>
> While multiple-choice questions (MCQs) are the standard verification method in current medical language reasoning models (e.g., MedQA, Huatuo-o1, MedReason), we agree that evaluating open-ended generation and pure reasoning capabilities is crucial.
>
> As shown in Table 5, to demonstrate that our Med-REFL framework enhances generic, open-ended reasoning capabilities beyond medical MCQs, we conducted additional experiments on the Knights and Knaves (K&K) logical puzzle dataset[1]. This task requires complex, multi-step logical deduction and is strictly open-ended (generation-based).
>
> We applied the Med-REFL framework ( prompt changed for logic puzzle) to the K&K training set (n=2,100 samples) to construct a `Puzzle-REFL` dataset. As shown in the table below, the framework demonstrates robust scalability:
> - **Method Generalizability:** The `+Puzzle-REFL` model achieves substantial gains (Avg **+9.59%** over baseline and **+8.45%** on hardest n=4 problem), proving that our method effectively instills reasoning capabilities in open-ended tasks.
>
> - **Cross-Domain Robustness:** Even the medical-specific `+Med-REFL` model shows positive transfer (+2.98%) on this logical task, indicating improved general reasoning faculties.
>
> **Table 1: Performance on K&K**
> |**Difficulty (Number of People)**|**Llama3.1-8b**|**+Med-REFL**|**+Puzzle-REFL**|
> |---|---|---|---|
> |**n=2**|38.77|40.82|**45.73**|
> |**n=3**|22.88|29.23|**36.24**|
> |**n=4**|13.40|13.96|**21.85**|
> |**Avg**|25.02|28.00|**34.61**|
>
> # A Process Evaluation Experiment ( Table 4, Line 393 )
>
> To address the concern regarding the lack of standard "process metrics" (e.g., faithfulness, coherence), we introduced a **mechanistic quantitative analysis** in Table 4( Line 393 ) that evaluates the "functional utility" of the reasoning process rather than just linguistic quality.
>
> We "opened the black box" by explicitly comparing the intermediate conclusions derived from the initial ‘Thinking’ phase ($A_{think}$) against the final ‘Critical Reflection’ phase ($A_{reflect}$). Instead of static metrics, we defined two dynamic process indicators:
>
> 1. **Reflection Divergence Rate:** This measures how often the model alters its decision after reflecting. Med-REFL increased this from 1.38% to 2.23%, indicating the model overcomes reasoning inertia and engages in active self-scrutiny.
>
> 2. **Correction Success Rate:** This measures the conditional probability that a change in answer successfully rectifies an error. This metric nearly doubled from 12.76% to 19.83%.
>
>
> This analysis proves that the generated reflection is not merely coherent text or stochastic noise; rather, the model has internalized a genuine **discriminative ability** to identify logical fallacies and execute valid self-corrections. We argue this functional validation provides a more rigorous evaluation of reasoning efficacy than traditional text-based metrics.

---

> ### Author Response · Authors · 2025-11-21
> **Response to Q4 & W3     [3/3]**
>
> # The Necessity of Reflection Text
>
> "Path preference alone" corresponds to our "Reasoning Enhancement" data (comparing valid vs. flawed CoT without explicit reflection). Our ablation study in Table 13 (Line 1303 ) explicitly compares this against "Reflection Learning" (which includes the reflection text).
>
> - **Empirical Superiority:** For the specialized model **Huatuo-o1**, training with Reflection data alone yielded a **+1.34%** gain, significantly outperforming Reasoning data alone (+0.57%). This indicates that for models already capable of basic reasoning, explicit reflection text provides a critical "cognitive scaffold" for error correction that simple path preference cannot offer.
>
> - **Internalized Capability:** Furthermore, the reflection text is necessary to "activate" the model's latent checking mechanism. As shown in Table 3 (Line 367), models trained with reflection text show a **+2.41%** surge in performance when prompted to "Reflect after thinking," proving they internalized the _logic_ of reflection, not just a probability distribution over paths.
>
>
> # Response to Weakness: Fixed λ weights and 1:2 data mix lack broader sensitivity studies.
> We have included detailed sensitivity analyses in Table 11 (Line 1217) and Figure6 (Line1559) which empirically validate our choices rather than relying on arbitrary fixation.
>
> - Sensitivity of Data Mix Ratio: We investigated the interplay between general reasoning and reflection training by varying the mix ratio from 1:1 to 1:4 on the MedQA dev set (Table 11).
> - Sensitivity of $\lambda$ Weights: It has been discussed in "Answer for Sensitivity Concern (Q1)"
>
> # Compute and Time Costs (Table6 , Line 433)
>
> **Table: Cost comparison between different reasoning enhancement paradigms. Estimated costs are based on standard A100 cloud rental rates (≈$2/hr) or public API pricing.**
> | Method | Resource Dependency | Est. Time / Cost |
> | :--- | :--- | :--- |
> | Distillation (GPT-5) | Paid API | >$2,500 (Token cost) |
> | Training PRM | Reward Model | >100 GPU Hours |
> | **Med-REFL (Data Gen)** | **Self-Generated** | **≈83 GPU Hours (≈$166)** |
> | **Med-REFL (Training)** | **DPO** | **≈12 GPU Hours** |
>
>
> We've compiled our GPU usage statistics, which are presented in this table. We spent approximately 83 hours on data generation, with ToT-Rollout accounting for about 70 hours, error location about 3 hours, and reflection generation about 10 hours. Compared to PRM or distillation expert model methods, our approach is cost-friendly.

---

> > ### Comment · Reviewer_dN7n · 2025-11-26
> >
> > Thank you for the detailed and thoughtful response, which has clarified many of my earlier questions. While I appreciate the effort, I remain somewhat unconvinced that the work introduces a substantially novel technical direction, as the contribution appears to be more incremental. The paper could be strengthened by highlighting any new phenomena, observations, or limitations identified during implementation, which may inspire future research. In addition, presenting motivations that go beyond improvements on benchmark scores and validate them, such as broader implications or insights into model behavior, would help convey the value and impact of the approach more effectively.

---

> ### Author Response · Authors · 2025-11-26
> **Beyond Benchmarks: Response Regarding Novelty and insight**
>
> Dear Reviewer dN7n,
>
> We sincerely appreciate your engagement and your insightful suggestion to look beyond benchmark scores. Prompted by your feedback, we re-examined our results through the lens of model behavior perspective.
>
> We believe the fundamental novelty of Med-REFL lies in solving a **critical scientific bottleneck** in current reasoning models: the prevalence of "fake" or "confirmatory" reflection. Our work does not merely improve accuracy; it improves how models utilize reflection.
>
> We highlight three key insights that validate this claim:
>
> 1. **The Background: Med-REFL Solves the "Fake Reflection" Crisis Identified by Concurrent Research**
>    There is a growing consensus that current reasoning models struggle to perform genuine self-correction. Some previous works have mentioned that they observed this phenomenon[1][2]. A concurrent ICLR 2026 submission, "First Try Matters[3]" (which received high recognition), conducts a large-scale analysis of 8 reasoning models. It reveals a harsh appearance:
>    - Reflections in current models are predominantly **confirmatory** (>90%) rather than corrective.
>     - The probability of a model correcting an initial error ($F \rightarrow T$) is negligible ($<2\%$).
>
>     **In contrast, Med-REFL provides a systematic solution to the 'Confirmatory Bias' problem** By constructing preference pairs that explicitly contrast "Effective Reflection" ($F \rightarrow T$) against "Invalid Reflection" ($F \rightarrow F$), we teach the model to learn the trajectory of correction.
>
> 2. **Evidence of Functional Shift: From "Stochastic Noise" to "Effective Correction"**
>
>    This claim is validated by our **process-level analysis in Table 4**, which directly addresses the limitation cited above.
>    **The Observation:** In the baseline model, changes in answers during reflection are often stochastic. However, Med-REFL significantly boosts the **Correction Success Rate** (the conditional probability that a change rectifies an error) from **12.76% to 19.83%** by introduce **DPO** to distinguish between effective reflection and fake reflection. This proves that Med-REFL effectively converts the "confirmatory" behavior into "corrective" behavior. **Table 3** also shows that Med-REFL encodes a latent self-correction mechanism that can be activated on demand
>
>
> 3. **Why use medical benchmark in main paper?** We focused on the medical domain in the main text because it serves as the most rigorous situation because of expensive cost in verifying reasoning steps. Med-REFL is a low-cost approach to address this issue. Besides, our methodology is domain-agnostic. As shown in our **Logic Puzzle experiment (Table 5)** , Med-REFL improves performance on open-ended logic tasks by **+9.59%** (avg).
>
> In summary, Med-REFL's contribution is not incremental; it provides a low-cost solution to the "ineffective reflection" problem that the community is currently grappling with. We hope these behavioral insights demonstrate the work's broader value.
>
> Best regards,
>
> Authors
>
> **Reference**
>
> [1] Zhang, Xiaoying, et al. "Critique-grpo: Advancing llm reasoning with natural language and numerical feedback." _arXiv preprint arXiv:2506.03106_ (2025)
>
> [2] Dang, Renfei, Shujian Huang, and Jiajun Chen. "Internal Bias in Reasoning Models leads to Overthinking." _arXiv preprint arXiv:2505.16448_ (2025).
>
> [3] Kang, Liwei, et al. "First Try Matters: Revisiting the Role of Reflection in Reasoning Models." _arXiv preprint arXiv:2510.08308_ (2025).

---

> > ### Comment · Reviewer_dN7n · 2025-11-27
> >
> > Thanks for your efforts. Your response has alleviated my concerns to a certain extent, and I have accordingly raised my score. I would strongly recommend that, in the published version, the relevant insights be articulated with more rigorous academic justification. While the work may give the impression of presenting similar experiments in a different narrative, the underlying motivation is distinct. The community does not need another score‑driven study, but rather contributions that offer clear perspectives or reveal novel phenomena, thus reducing, rather than increasing, entropy in the discourse. Overall, I appreciate the authors’ efforts and retain a positive view of the work.

---

### Author Response · Authors · 2025-12-01
**Summary of Revision**

**Dear reviewers and AC,**

We are sincerely grateful for the valuable time you dedicated to reviewing our paper and offering insightful feedback. We have carefully revised the paper in line with your comments and highlighted the modifications in **blue** within the updated manuscript. A summary of the key revisions is provided below:

* **Addressed "Fake Reflection" and Confirmatory Bias:** We added a dedicated discussion on the "fake reflection" phenomenon and confirmatory bias in **Section B.4** (Appendix) and **Section 4.2** to address community concerns regarding the validity of self-correction (Reviewer dN7n).
* **Quantitative Analysis of Intermediate Process:** We included a mechanistic analysis in **Section 4.2 (Table 4)**. This analysis quantifies the *Reflection Divergence Rate* and *Correction Success Rate*, demonstrating that Med-REFL effectively mitigates the "fake reflection" phenomenon in LRMs (Reviewer dN7n).
* **Cross-Domain Generalization:** We conducted additional experiments on the **K&K Logic Puzzle Benchmark** to demonstrate the method's generalization capabilities beyond the medical domain. These results are presented in **Section 4.3 (Table 5)** (Reviewer dN7n & oh3a).
* **Cost and Efficiency Analysis:** We added a detailed analysis of data construction and training costs in **Section 4.4 (Table 6)**, highlighting the efficiency of our framework compared to PRM and distillation methods (Reviewer dN7n, oh3a & 3fRk).
* **New Discussion Section:** We introduced a **Discussion module (Section 5)** to analyze intriguing phenomena observed during experiments, addressing specific reviewer inquiries:
    * Domain sensitivity and parameter adjustments (Reviewer 3fRk).
    * The "Reflection as Attention Redirection" paradigm (Reviewer 83j9).
    * Mechanisms for mitigating fake reflection via contrastive learning (Reviewer dN7n).
* **Sensitivity and Robustness Analysis:** We added a comprehensive sensitivity analysis in **Appendix F.9**, examining the impact of sampling parameters, search budgets, and action value weights from a reasoning boundary perspective (Reviewer dN7n, 83j9 & oh3a).
* **Reorganization for Clarity:**
    * To improve flow, we moved the *Scoring Model Analysis* (**Figure 3**) and the *DPO vs. SFT Ablation Study* (**Table 12**) to **Appendix F.3** and **Appendix F.4**, respectively.
    * We added **Section 4.6** in the main text to provide a clear index and summary of the extensive ablation studies and validations located in the Appendix.

We genuinely hope these revisions have adequately addressed your concerns.

Best,

Authors

---

### Author Response · Authors · 2025-12-01
**[Summary to AC] Rebuttal Status Update: Consensus with Reviewer dN7n and Other Responses**

Dear Area Chair,

We understand the significant challenges posed by the recent OpenReview incident and the additional workload required to re-evaluate submissions. As per the program chairs' guidance, we are posting this summary to assist you in estimating **"how the reviewer's impressions would have changed"** based on our rebuttal and the discussions that took place before the freeze.

Specifically, we would like to draw your attention to the **substantive and successful discussion with Reviewer dN7n (Confidence: 4)**, which concluded with a score increase **before** the widespread incident announcement.

### 1. **Resolution of Concerns with Reviewer dN7n ( Score raised 4 -> 6 )**

Reviewer dN7n was the most engaged reviewer. Following our initial rebuttal and a second round of in-depth discussion, the reviewer explicitly acknowledged the value of our work and raised their score.

- **Initial Concern:** The reviewer initially questioned the technical novelty, even after we had already submitted our first round of experimental rebuttal.

- **Resolution:**  In our second response (Nov 26), we provided mechanistic analysis and cited concurrent works (e.g., "First Try Matters" and "Internal bias") to demonstrate that our method solves the specific "fake reflection" phenomenon in reasoning models. This is another important contribution of our work to the community.

- **Reviewer's Final Feedback (Nov 27, 12:52 UTC):**  Reviewer dN7n acknowledged this distinct motivation, stating: _" I have accordingly raised my score... The underlying motivation is distinct... I appreciate the authors’ efforts and retain a positive view of the work,"_ and subsequently raised the score.

- **Outcome:** The reviewer raised their rating to **6** .


### 2. **Status of Other Reviewers**

We have submitted a revised PDF (Revision 2) and detailed responses to all other reviewers (Reviewer 83j9, oh3a, and 3fRk), addressing their concerns regarding experimental details and comparisons. Unfortunately, the discussion period was cut short before they could respond. However, given that Reviewer dN7n (who had the **highest confidence** and **most critical initial stance**) was satisfied by our additional analyses and clarifications, we remain optimistic that our responses would have likely **positively influenced** the other reviewers' final assessments as well.

### 3. **Conclusion**

The consensus reached with Reviewer dN7n demonstrates that Med-REFL's value lies in its unique capability to internalize genuine self-correction, moving beyond simple score-driven metrics. In line with Reviewer dN7n’s recommendation, we incorporated a new **Discussion** section (Section 5) to explore intriguing phenomena observed during implementation, aiming to provide the community with deeper insights into model behavior rather than just benchmark improvements.

**We respectfully request that you consider this completed and positive interaction as a key signal of the paper's quality during your evaluation.**

Thank you for your time and dedication to ensuring a fair review process under these difficult circumstances.

Sincerely,

Authors

---

### Meta-Review · Area_Chair_HypR · 2026-01-03

**Summary:**

This paper provides a rather sophisticated framework to instill fine-grained self-correction in medical LLMs. The reviewers raised several weaknesses:

1) Sensitivity analysis of ToT parameters and tunable lambda weights. Performance may depend on the quality of LLM-based tools and hyperparameter choices.

2) No study on open-ended tests. Metrics on faithfulness, hallucination rates, calibration, etc.

3) DPO training pairs are mainly from MedQA and are perhaps narrow.

4) The paper lacks novelty, and the contribution is incremental.

5) Limited algorithmic contribution. It is more like porting self-reflection techniques to the medical domain.

6) Mixed results on retrieval-oriented datasets.

7) Computational cost is too high. What if the data is 10X? Can the system scale to that level?

8) In Table 1, the authors claimed that their method is “PRM-free.” However, the authors still use a small-scale reward model.

9) Failure analysis.

**Reviewer Concerns:**

1) The authors report added sensitivity analyses and robustness to ToT hyperparameters and action-value weights. However, they did not answer how different LLM tools (error locator and reflector) may affect the performance. I also feel the idea of using lambdas to combine different values in the tree algorithm (Eqn (4)) is quite ad-hoc.

2) Additional experiments on the Knights and Knaves (K&K) logical puzzle dataset. However, I don’t believe the authors have answered the question of faithfulness, hallucination rates, and calibration.

3) The authors use the K&K dataset results to show that the Med-REFL framework improves generic, open-ended reasoning capabilities beyond medical MCQs. However, I don’t think that addresses the key concern that “preference data sourced mainly from MedQA”.

4) The authors highlighted the key novelty in addressing "fake" or "confirmatory" reflection. This framing helps, but whether it is a substantial technical novelty depends on how clearly the paper demonstrates these points.

5) The authors emphasized that their method provides” deterministic, structure-based evaluation to automatically synthesize a high-quality preference signal” to improve self-reflection. While the intent is clear, the evidence that this signal is “high-quality” beyond empirical gains is limited. The metrics introduced from (1)-(4) seem rather ad-hoc, and there are three parameters to tune in eqn (4).

6) The authors provided some possibilities and promised to explore further.

7) The authors claimed that the cost is mainly for offline data generation, while the online prediction is not affected (because no additional inference module is added). They also provided the time cost in Table 6. Regarding the scalability issue, the authors claimed that on a new logic-puzzle task, one can avoid using a large amount of initial data. However, the rebuttal does not directly answer the core scalability question.

8) I believe the authors’ claims are not substantiated. Even if the reward model is lightweight, it is still a reward model.

9) The authors provided extra failure mode analysis.

**Reviewer Scores:**

If the reviewers engage fully, the scores could increase from 4464 to 6464, or even slightly higher. However, I don't the authors have fully addressed the key limitations outlined in the meta review above.

---

### Decision · Program_Chairs · 2026-01-26

Reject